# A Comprehensive Review on Weight Loss Associated with Anti-Diabetic Medications

**DOI:** 10.3390/life13041012

**Published:** 2023-04-14

**Authors:** Fatma Haddad, Ghadeer Dokmak, Maryam Bader, Rafik Karaman

**Affiliations:** 1Pharmaceutical Sciences Department, Faculty of Pharmacy, Al-Quds University, Jerusalem 9103401, Palestine; iamfromhebron@hotmail.com (F.H.); ghadeer_88@live.com (G.D.); mariam407@hotmail.com (M.B.); 2Faculty of Life Sciences, University of Bradford, Bradford BD7 1DP, UK; 3Department of Sciences, University of Basilicata, 85100 Potenza, Italy

**Keywords:** obesity, T2DM, GLP-1 agonist, weight, loss

## Abstract

Obesity is a complex metabolic condition that can have a negative impact on one’s health and even result in mortality. The management of obesity has been addressed in a number of ways, including lifestyle changes, medication using appetite suppressants and thermogenics, and bariatric surgery for individuals who are severely obese. Liraglutide and semaglutide are two of the five Food and Drug Administration (FDA)-approved anti-obesity drugs that are FDA-approved agents for the treatment of type 2 diabetes mellitus (T2DM) patients. In order to highlight the positive effects of these drugs as anti-obesity treatments, we analyzed the weight loss effects of T2DM agents that have demonstrated weight loss effects in this study by evaluating clinical studies that were published for each agent. Many clinical studies have revealed that some antihyperglycemic medications can help people lose weight, while others either cause weight gain or neutral results. Acarbose has mild weight loss effects and metformin and sodium-dependent glucose cotransporter proteins-2 (SGLT-2) inhibitors have modest weight loss effects; however, some glucagon-like peptide-1 (GLP-1) receptor agonists had the greatest impact on weight loss. Dipeptidyl peptidase 4 (DPP-4) inhibitors showed a neutral or mild weight loss effect. To sum up, some of the GLP-1 agonist drugs show promise as weight-loss treatments.

## 1. Introduction

According to the World Health Organization (WHO), diabetes mellitus is a long-term metabolic condition marked by hyperglycemia that over time can harm the heart, blood vessels, eyes, kidneys, and nerves [1,2]. Type 2 diabetes mellitus (T2DM) affects more than 90% of individuals living with diabetes mellitus [1]. T2DM is a heterogeneous, progressive metabolic condition defined by tissue insulin resistance, insufficient compensatory insulin production, and faulty insulin secretion by pancreatic beta-cells [1]. Due to an increase in sedentary behaviors, obesity, and poor dietary choices around the world between 1980 and 2004, the incidence and prevalence of T2DM quadrupled [3]. The principal treatments for T2DM patients include insulin secretagogues, biguanides, insulin sensitizers, alpha-glucosidase inhibitors, incretin mimetics, amylin antagonists, and sodium-dependent glucose cotransporter proteins-2 (SGLT-2) inhibitors [4]. Pharmacological agents may be used either as monotherapy or with other hypoglycemic drugs [5]. Adapting the present medicines to produce optimal and balanced glucose levels and reducing long-term complications related to diabetes are the major obstacles to effective diabetes control, despite the introduction of promising anti-hyperglycemic drugs [5,6].

The term “diabesity” captures the intimate connection between the twin epidemics of diabetes and obesity [7]. Obesity is a multifactorial metabolic syndrome that raises the risk of several illnesses, including T2DM, cardiometabolic disorders, dementia, depression, and malignancies, which can lower quality of life and result in death [8,9]. Over the past 50 years, the prevalence of obesity has increased to epidemic levels worldwide [8]. Improvements in lifestyle habits, including strategies to improve diet and exercise quality, pharmacotherapy, such as appetite suppressants and thermogenic agents, and bariatric surgery in patients who are extremely obese have all been proposed as ways to control obesity [10,11]. Currently, orlistat, phentermine-topiramate, naltrexone-bupropion, liraglutide, and semaglutide have all been approved by the Food and Drug Administration (FDA) for long-term use in the treatment of obesity [12]. Liraglutide and semaglutide are two of the antihyperglycemic medications that are utilized and have FDA approval for the treatment of people with type 2 diabetes [13,14].

Several studies have demonstrated that some antihyperglycemic medications can help T2DM patients lose weight, whereas other antihyperglycemic medications are weight-neutral [15]. Sulfonylureas, thiazolidinediones, and insulin, on the other hand, caused weight gain [15].

In order to highlight the promising effects of these medications as anti-obesity treatments, this study analyzed T2DM medications that have the potential to reduce weight by examining clinical trials that were published on each agent involving weight reduction effects.

## 2. Method

An electronic search of the literature was conducted using the following databases: Google Scholar; PubMed; Scopus; and Science Direct, and the following keywords: anti-diabetic medications; T2DM agents; obesity; anti-obesity treatments; weight; loss; BMI; biguanide; metformin; alpha-glucosidase inhibitors; acarbose; T2DM; DPP-4; SGLT-2; GLP-1 agonist; sitagliptin; vildagliptin; alogliptin; saxagliptin; linagliptin; canagliflozin; dapagliflozin; empagliflozin; ertugliflozin; exenatide; liraglutide; dulaglutide; lixisenatide; semaglutide; and tirzepatide. The most recent research that met the following selection criteria were included in this review: (1) written in English; (2) reported weight change associated with anti-diabetic medications either as primary or secondary endpoint; (3) analyzed T2DM agents that can potentially reduce weight; and (4) examined the effects on subjects with or without diabetes or with or without obesity. Studies that have not explored the weight effect of T2DM agents on subjects were excluded. Moreover, studies that have examined T2DM agents which cause weight gain such as sulfonylureas, thiazolidinediones, and insulin were also excluded [1]. According to this study, weight loss that was less than about 3% of the starting weight was deemed mild, between about 3% and 5% was deemed moderate, and more than 5% reduction was deemed strong, as was the case in the Lazzaroni et al. study [2].

## 3. Result and Discussion

### 3.1. T2DM Agents That Have the Potential to Reduce Weight

#### 3.1.1. Biguanide

Biguanide belongs to the family of guanidine, which is derived from Galega officinalis, a French lilac that has been used for its anti-diabetic, anti-hypertensive, and anti-aging properties since the Middle Ages [16,17]. Its anti-cancer activities were also demonstrated by preclinical and clinical investigations [18]. With the discovery of galegine at the beginning of the 20th century, many biguanides (such as synthelin A and B, biguanide, metformin, phenformin, and buformin) were developed and investigated as antidiabetic medications [18]. Metformin (1,1 dimethylbiguanide) is arguably the most well-known biguanide [17], while other biguanides were abandoned because of toxicity issues or a perception of low potency [18].

##### Metformin

Because of its exceptional effectiveness, safety profile, tolerance, and lack of hypoglycemia, metformin (Figure 1a), an antihyperglycemic biguanide, is well-known as a cornerstone and first-line treatment of T2DM that decrease hepatic glucose production (Figure 2) [19,20]. Current research has demonstrated that metformin offers additional therapeutic advantages than lowering glycemia, such as extending life, reducing body weight, and decreasing the risk of developing cancer [21]. Moreover, the only biguanide medication that has received FDA approval for the treatment of hyperglycemia in T2DM patients is metformin [17]. Metformin mostly causes weight loss by lowering hunger and resulting in consuming less calories [21]. Both the medication’s direct cerebral action and its gastrointestinal side effects, such as nausea, bloating, diarrhea, and dysgeusia, indirectly affect how well it controls appetite [21].

The Diabetes Prevention Program (DPP) is the largest program examining the advantages of metformin for weight loss [22]. The DPP was a randomized controlled trial that compared weight loss with metformin, intensive lifestyle treatments, or placebo and evaluated the preventative effects of the drug on metabolic parameters in individuals at high risk for T2DM [22,23]. A total of 3234 people participated in the program at random. Those who lost at least 5% of their starting body weight underwent surveillance for more than 15 years, according to the DPP observational study [23]. After 15 years, the metformin group’s average weight loss was 6.2% as opposed to the lifestyle group’s 3.7% [23]. Moreover, 56% of patients in the metformin group kept their weight loss at least at 5%, as opposed to 43% of patients in the lifestyle group [23]. In total, 86 people with newly diagnosed T2DM were randomly assigned to receive either gliclazide, metformin, or acarbose for 6 months by Wang et al. [24]. According to the study, a patient’s body fat mass (3.51 kg) and body fat percentage (4.45%) decreased statistically significantly after 6 months of treatment [24]. When compared to other patients, those using metformin had significant weight loss, going from 71.6 kg to 68.4 kg [24]. From 6.45 kg to 6.12 kg, the metformin group’s total body fat decreased by 4.5% [24]. Additionally, given that the majority of antipsychotic medications produce weight gain, metformin’s impact on weight gain brought on by these medications has been studied in numerous randomized trials [25]. When metformin was combined with atypical antipsychotics, there was a significant reduction in body mass index (BMI) and insulin resistance without changing fasting blood sugar, according to a recent meta-analysis of 12 studies that included a total of 743 individuals [25]. The average weight change was 3.27 kg, and metformin significantly decreased BMI [25]. Metformin’s ability to prevent weight gain brought on by antipsychotics in individuals with schizophrenia or schizoaffective disorder was supported by the meta-analysis [25].

A systematic review also gave an overview of the impact of metformin treatment for 6 months on weight, insulin resistance, and progression to T2DM in 14 adult trials and 15 pediatric studies [26]. Children experience lesser weight/BMI reductions on metformin than adults do [26]. Variations in adherence, dose, and insulin state could cause this [26]. Metformin significantly reduced the progression of T2DM in adults by 7–31%, but no average weight loss of more than 5% was found [26].

A total of 85 T2DM patients with non-alcoholic fatty liver disease participated in a 24-week randomized experiment to evaluate the effects of gliclazide, liraglutide, and metformin on body composition [27]. Compared to gliclazide, liraglutide and metformin monotherapies result in greater weight loss, lower body fat percentages, and better blood glucose management in T2DM patients [27]. The only groups who saw significant weight loss were those using liraglutide and metformin (from 81.1 ± 2.3 kg to 75.5 ± 2.0 kg, *p* < 0.01) and from 74.8 ± 2.5 kg to 71.2 ± 2.6 kg, *p* < 0.01, respectively) [27]. Metformin’s impact on weight loss was assessed in a recent meta-analysis of 21 trials including a total of 1004 participants [28]. In obese people with a BMI greater than 35 kg/m^2^, the experiment found that metformin treatment significantly reduced BMI as compared to baseline [28]. The BMI dropped by 1.01 units in the high-dose groups, although it did not continue to drop appreciably after 6 months [28]. To assess whether this reduced value resulted in enough weight loss (5% of baseline body weight) to qualify metformin as a weight loss medication, they argued that larger and more randomized control trials are required [28]. Another recent retrospective study evaluated the outcomes within 6–12 months of follow-up in 222 individuals (103 euglycemic patients and 119 T2DM/prediabetes patients) who finished metformin alone treatment for weight loss [29]. The average weight loss across the euglycemic and T2DM/prediabetes groups was nearly the same after six months, according to the results: 6.5 ± 6.0% (euglycemic) vs. 6.5 ± 6.1% (T2DM/prediabetes), *p* = 0.97. At one year: 7.4 ± 6.2% (euglycemic) vs. 7.3 ± 7.7% (T2DM/prediabetes), *p* = 0.92 [29]. They proposed that treating obesity with metformin and modifying one’s lifestyle could be successful for both patients with and without T2DM/prediabetes [29]. Hence, metformin should be considered when treating obesity, especially in regions with insufficient access to FDA-approved medications for long-term weight management [29].

#### 3.1.2. Alpha-Glucosidase Inhibitors

The alpha-glucosidase enzymes required for carbohydrate digestion in the small intestines are competitively inhibited by alpha-glucosidase inhibitors, which are saccharides (Figure 2) [30]. The enzymatic breakdown of carbohydrates into simple absorbable sugars is reversibly inhibited by them [30]. As a result, they inhibit the absorption of carbs and lessen the increase in postprandial blood glucose by around 3 mmol/L [30]. It has been proposed and proven that alpha-glucosidase inhibitors, and particularly acarbose, can be used to reduce body weight by blocking the absorption of carbs and therefore reducing the calorie intake [31,32]. Additionally, they proposed that increasing glucagon-like peptide-1 (GLP-1) production could be a potential mechanism for the weight loss associated with alpha-glucosidase inhibitor (acarbose) [33]. The effectiveness and safety of alpha glucosidase inhibitors, including acarbose, were assessed using data from 67 trials conducted worldwide [32]. They came to the conclusion that these drugs considerably lowered body weight when compared to a placebo, without raising the risk of hypoglycemia, but did so at the expense of an increased incidence of gastrointestinal discomfort [32].

##### Acarbose

The FDA has approved acarbose (Figure 1b), an alpha-glucosidase inhibitor, for the treatment of T2DM patients either alone or in combination with other antidiabetic medications [34]. Moreover, acarbose has been shown to lengthen the lives of T2DM patients and lessen their chance of developing cardiovascular diseases [35,36,37]. Acarbose was found to cause weight loss in several studies [38,39]. Acarbose’s impact on body weight was investigated in a global, non-interventional, observational trial that aggregated 10 post-marketing studies (*n* = 67,682 participants) [37]. With an average weight loss of 1.45 ± 3.24% (*n* = 43,510; mean body weight baseline 73.4 kg) at the 3-month visit and 1.40 ± 3.28% (*n* = 54,760; mean body weight baseline 73.6 kg) at the final visit, a significant body weight reduction was seen [37]. According to their findings, acarbose significantly reduced body weight in T2DM patients which is independent of glycemic control in these patients [37]. This benefit is contingent on baseline body weight, though, since patients with higher baseline body weights and/or BMIs experienced greater weight loss [37]. Acarbose and metformin were compared for 48 weeks in a randomized, open-label experiment to see which was more effective [38]. A total of 784 T2DM patients started their medication trials, of which 393 were given metformin and 391 were given acarbose [38]. For acarbose and metformin, the average weight loss was 2.55 kg and 1.88 kg, respectively [38]. Surprisingly, acarbose had a greater impact on weight loss than metformin [38]. In a randomized clinical trial, the variables influencing the waist-to-height ratio, a measure of abdominal obesity, when acarbose or metformin were used as monotherapy for 24 weeks were examined [39]. A total of 343 T2DM participants were assigned to the acarbose group and 333 were assigned to the metformin arm [39]. Both treatment groups showed a considerable decline in the waist-to-height ratio [39]. The individuals were divided into 2 groups based on changes in the waist-to-height ratio at week 24 using the median as the cutoff (−0.012): high differences in the waist-to-height ratio group and low differences in the waist-to-height ratio [39]. The patients who received acarbose and had significant variations in their waist-to-height ratios experienced average body weight losses of 3.63 kg (5.28%) [39]. While those with a low waist-to-height ratio dropped an average of 1.29 kg (1.84%) of body weight [39]. Their results proved that the increase in GLP-1 levels brought on by acarbose is what causes this reduction in the acarbose group [39]. Nonetheless, acarbose’s impact on individuals who were overweight and obese but did not have diabetes was recently investigated in a meta-analysis [40]. Their systemic evaluation contains a total of seven randomized controlled studies [40]. In total, 5 of them were chosen to evaluate acarbose’s impact on BMI, which included 84 participants in the control group and 80 people in the acarbose arm [40]. In the two groups, there was no clinically significant variation in BMI [40]. However, their sensitivity analysis revealed that the outcome was unstable, a significant reduction of BMI in the acarbose group was observed (−1.82), which could be related to the drug dose and the brief treatment period [40]. To evaluate the effects of acarbose compared with the dipeptidyl peptidase-4 (DPP-4) inhibitor, a comprehensive review and network meta-analysis of randomized controlled trials were conducted [41]. Their meta-analysis and network meta-analysis, which included 13 pair-wise studies and 48 monotherapy trials, respectively [41]. The pair-wise meta-analysis findings revealed that DPP-4 had noticeably greater impact on the control of hyperglycemia [41]. However, acarbose and DPP-4 were found to have comparable effects on the control of hyperglycemia and HbA1C in a network meta-analysis involving 11,877 T2DM patients [41]. When employing the best dosages, acarbose, as opposed to DPP-4, had an interestingly superior impact on weight loss in such patients [41]. EMP 16 is a brand-new weight-loss combination product that was created as a prospective weight-loss product containing orlistat and acarbose [42]. A six-month, randomized, double-blind, placebo-controlled research study was conducted in 2022, employing 156 obese subjects to examine the impact of this unique combination on weight loss [42]. As comparison to the placebo group, the participants who were given EMP 16 demonstrated a 5% greater reduction in body weight [42]. Longer trials are still required to assess the efficacy and safety of this unique combination, despite the fact that it may be a potential candidate for weight loss [42]. In conclusion, acarbose has mild effects on weight loss in T2DM patients.

#### 3.1.3. DPP-4 Inhibitors

DPP-4 inhibitors are brand new oral medications that have the potential to be useful in treating T2DM [43]. Incretin hormones, primarily GLP-1 and gastric inhibitory peptide (GIP), which control insulin and glucagon secretion to maintain glucose homeostasis, are influenced by the widely distributed enzyme DPP-4 [43,44]. GLP-1 and GIP are not rendered inactive by DPP-4 inhibitors (Figure 2). As a result, it has an impact on glucose regulation by raising GLP-1 levels, which increase insulin secretion and decrease glucagon secretion without resulting in intrinsic hypoglycemia [43,44]. These medications can reduce hemoglobin A1c (HbA1c) levels by 0.5% to 1.0%, according to several studies [43]. In addition to their hypoglycemic effects, they also have non-incretin pathway dependent antihypertensive, anti-inflammatory, anti-apoptotic, and immunomodulatory effects [45,46]. There are now five DPP-4 inhibitors on the market that have received regulatory approval: sitagliptin, vildagliptin, alogliptin, saxagliptin, and linagliptin [44].

##### Sitagliptin

The FDA authorized sitagliptin (Figure 1c) in 2006 as the first oral DPP-4 inhibitor [47]. Sitagliptin is an oral diabetic medication that improves glycaemic control while having a low incidence of hypoglycemia in persons with T2DM of all ages [47]. It has a weight-neutral effect, is well-tolerated, and is only modestly effective [47]. It might have a special therapeutic impact on diabetics with liver or kidney problems [47]. Several research studies were done to assess how it affected T2DM patients’ ability to lose weight.

A total of 372 elderly T2DM patients over the age of 65 underwent a post-hoc analysis of three double-blinded randomized studies [48]. Together with weight loss, they contrasted the glycaemic impact of sitagliptin vs. sulfonylurea. Body weight was significantly reduced by sitagliptin by 1.7 kg, whereas it was significantly elevated by sulfonylurea by 0.5 kg [48]. The results of their investigation showed that sitagliptin had body weight loss effect, less hypoglycemia, and a similar glycemic effect to sulfonylurea on T2DM patients [48]. Moreover, sitagliptin was compared to sulfonylureas in a comprehensive review and meta-analysis of seven randomized controlled trials and five non-randomized studies to determine which was more effective as an add-on therapy to metformin in individuals with T2DM [49]. A meta-analysis of three homogenous randomized controlled studies (*n* = 1303, T2DM patients) revealed a statistically significant weight loss impact of sitagliptin when compared to sulfonylureas (weighted mean difference: −2.05 kg) [49]. The quality of three of the five non-randomized trials was either moderate or good [49]. The results of these three non-randomized investigations on both groups showed the same glycemic and weight effect [49]. Although the difference in weight loss between patients taking sitagliptin and those taking sulfonylureas was only about 2 kg, it may not have been clinically relevant for most participants [49]. In addition, 75 T2DM patients with non-alcoholic fatty liver disease (NAFLD) participated in a 26-week randomized experiment [50]. It looked at how adding liraglutide, sitagliptin, or insulin glargine to metformin affected body weight and intrahepatic lipid levels. Weight decrease from sitagliptin (from 88.2 ± 13.6 kg to 86.5 ± 13.2 kg) was considerable [50]. In addition to improving glycemic control in individuals with T2DM and NAFLD, the trial found that only liraglutide and sitagliptin added to metformin, but not insulin glargine, induced a reduction in body weight, visceral adipose tissue, and intrahepatic lipid [50].

On 2009 obese and overweight T2DM patients, a recent comprehensive review and meta-analysis of 18 randomized controlled studies investigated the impact of sitagliptin monotherapy as an add-on therapy to metformin on weight loss [51]. Hence, whether sitagliptin was given alone or in conjunction with metformin, weight reduction was caused [51]. The weight mean difference in patients receiving sitagliptin alone was −0.99 kg, while it was −1.09 kg in those receiving sitagliptin plus metformin [51]. They came to the conclusion that body weight may drop if sitagliptin is taken for longer than six months, whether or not it is combined with metformin [51]. In 11 different locations around Pakistan, a brand new prospective observational study involving 132 patients was conducted during Ramadan [52]. Before and throughout Ramadan, 88 patients participated in an investigation of the combination of metformin and sitagliptin’s effects on weight loss [52]. The group reported gains in weight and BMI both before and during Ramadan [52]. The data show that BMI changed from 39.5 ± 5.7 kg/m^2^ before Ramadan to 34.9 ± 4.8 kg/m^2^ after Ramadan [52]. Diabetes patients who fast throughout Ramadan may have weight loss while taking sitagliptin and metformin, and both medications are safe and well-tolerated [52]. According to most of the earlier research, sitagliptin, whether given either alone or in combination with other drugs, may result in mild weight loss.

##### Vildagliptin

Vildagliptin is an oral DPP-4 inhibitor and hypoglycemic drug (Figure 1d) [53]. It was created in 2007 and proved to be effective at lowering blood sugar without resulting in weight gain or increasing the risk of hypoglycemia [53]. Both when used alone and in combination with other anti-diabetic drugs or insulin, vildagliptin has proven to be successful [54]. Both monotherapy and combination therapy for T2DM are permitted to use it [55]. Vildagliptin elevates plasma levels of the intact incretin hormones GLP-1 and/or GIP by inhibiting the long-acting, competitive, and reversible enzyme DPP-4. By inhibiting unneeded alpha-cell glucagon secretion and promoting the growth of glucose-dependent beta cells, it consequently improves glucose homeostasis [56].

For 2340 T2DM patients who received vildagliptin monotherapy in 2014, a total of 8 randomized, controlled, double-blinded clinical monotherapy trials were conducted [57]. In comparison to baseline fasting plasma glucose (FPG) levels, they assessed how much weight changed over the course of the treatment (24 weeks) [57]. The average weight loss in the study population was 0.72 kg [57]. Simple linear regression was used to examine weight change after 24 weeks in relation to baseline FPG, and the findings showed an intercept of −2.259 kg and a positive slope of 0.1552 kg [57]. There was no change in weight with an FPG of 14.6 mmol/L (263 mg/dL) [57]. Weight loss and weight increase were associated with baseline FPG concentrations below and above this threshold, respectively [57]. For instance, a baseline FPG of 8 mmol/L forecasts a weight loss of 1 kg [57]. The current study proved that vildagliptin medication has a negative caloric balance when glucose levels are below the renal threshold at baseline [57]. Additionally, a recent systematic analysis of 8 journals clarified the effectiveness of vildagliptin in (*n* = 741) T2DM patients, both alone and in combination with metformin [58]. Vildagliptin and metformin have been proven to work well together to lower HbA1c, minimize the risk of hypoglycemia, and significantly reduce body weight, resulting in weight losses of 4.67 ± 5.8 kg for low dose combinations and 4.29 ± 6.7 kg for high dose combinations [58]. Their research found that vildagliptin was more effective in T2DM patients when combined with metformin than when used alone. Another recent meta-analysis examined the effectiveness and safety of combination therapy with vildagliptin and metformin vs. metformin alone for weight loss in 11 randomized controlled studies involving a total of 8533 T2DM patients [59]. Vildagliptin and metformin were used in combination therapy, which reduced the body weight loss ratio by 0.22 when compared to metformin monotherapy [59]. The results of the trial showed that, in comparison to metformin alone, the combination of vildagliptin and metformin significantly decreased FPG, HbA1c, and body weight [59]. In conclusion, vildagliptin showed mild effects when combined with metformin and neutral effects when administered alone.

##### Saxagliptin

At doses of 2.5 mg or 5 mg once daily, saxagliptin (Figure 1e) is a DPP-4 inhibitor that is authorized to treat T2DM [60]. Orally ingested saxagliptin is a very potent, selective, and competitive inhibitor of the DPP-4 enzyme [60]. Compared to vildagliptin and sitagliptin, its potency is 10 times more [60]. Much research has shown how it affects weight. A 24-week multicenter, double-blinded, randomized, and controlled phase 3 study evaluated the safety and effectiveness of the combination therapy of saxagliptin and dapagliflozin compared to a single addition of saxagliptin and dapagliflozin to metformin in T2DM patients (*n* = 1282) who are not well controlled with metformin alone [61]. The combination of saxagliptin and dapagliflozin led to a mean weight loss of 2.1 kg, as opposed to 2.4 kg with dapagliflozin alone and 0 kg with saxagliptin monotherapy [61]. Additionally, a 52-week multicenter, randomized, double-blinded, parallel-group experiment compared the effects of dapagliflozin plus saxagliptin and metformin to glimepiride and metformin in (*n* = 82) T2DM patients [62]. The dapagliflozin + saxagliptin + metformin group’s patients’ body weight decreased from 90.8 ± 19.7 kg at baseline to 88.4 ± 18.1 kg at week 52, according to the results [62]. In total, 24 obese patients with impaired glucose tolerance were evaluated for their glucose tolerance and the effects of saxagliptin treatment on blood glucose levels during fasting and postprandially after a typical breakfast in a randomized, placebo-controlled, double-blinded, controlled phase 2 study [63]. Between visits 1 and 2, both saxagliptin and placebo groups lost weight [63]. There were no statistically significant differences between the two groups’ waist measurements and body weight changes. Weights decreased for saxagliptin and placebo, respectively, from 107.5 ± 18.5 kg to 106.1 ± 20.2 kg and from 97.2 ± 14.5 kg to 93.5 ± 16.6 kg [63]. Waist circumference dropped from 114 cm to 109 cm in the saxagliptin group and from 108 cm to 102 cm in the placebo group [63]. In total, 648 Chinese patients with newly diagnosed T2DM were randomly assigned 1:1:1 to receive saxagliptin combination with metformin, acarbose, or gliclazide modified release tablets in a 24-week, multicenter, controlled research [64]. saxagliptin + metformin and Saxagliptin + acarbose groups’ body weights decreased by 1.6 kg and 1.5 kg, respectively, at week 24, but the saxagliptin + gliclazide group’s weight increased by 1.0 kg [64]. As stated, saxagliptin, when administered alone has a weight-neutral effect on T2DM patients’ weight, or may even produce weight loss when administered in conjunction with metformin

##### Linagliptin

With a minimal risk of hypoglycemia and weight neutrality, linagliptin (Figure 1f) is a once-daily selective DPP-4 inhibitor that has been approved for use in the treatment of T2DM [65,66].

In a single center parallel double-blinded trial with a 24-month follow-up, patients with impaired glucose tolerance and two T2DM risk factors were randomly assigned to receive either metformin 1700 mg plus linagliptin 5 mg daily plus lifestyle (group A, *n* = 70), or metformin 1700 mg plus lifestyle (group B, *n* = 74) [67]. At 24 months, both groups experienced a consistent and considerable decline in body weight and BMI [67]. With no statistically significant differences (*p* > 0.1) between the two groups, Group A had lost 4.1 kg, while Group B had lost 4.3 kg [67]. Additionally, the prospective, randomized controlled trial indicated that linagliptin had a weight-neutral impact [68]. In total, 206 Chinese patients with poorly controlled T2DM were given linagliptin as an adjuvant therapy to basal or premixed insulin, either alone or in combination with metformin [68]. Linagliptin 5 mg/day was given to subjects in a 1:1 ratio, or a placebo [68]. According to research, adding linagliptin to insulin therapy, either as monotherapy or in combination with metformin, significantly improved glycemic control and was well-tolerated in Chinese patients with T2DM, with no increased risk for hypoglycemia or weight gain compared to placebo [68]. In patients with persistent impaired glucose tolerance after a year of metformin and lifestyle adjustments, the effects of adding linagliptin to metformin and lifestyle changes on glucose levels and pancreatic beta-cell function were investigated [69]. Participants were randomized to receive either 2.5 mg of linagliptin plus 850 mg of metformin twice daily (group B, *n* = 19) or 850 mg of metformin twice daily (group A, *n* = 12) [69]. According to the results, patients in group B experienced weight loss (1.7 ± 0.6 kg, *p* < 0.05), as well as decrease in BMI (0.6 ± 0.2 kg/cm^2^, *p* < 0.05), and their waist circumference (3.6 ± 1.5 cm, *p* < 0.05) [69]. When compared to group A, these differences, however, were not clinically significant [69]. Additionally, 106 Asian individuals with T2DM receiving premixed insulin participated in a 24-week open-label, randomized research that examined the effects of 25 mg of empagliflozin and 5 mg of linagliptin on body weight and body composition [70]. In the linagliptin and empagliflozin groups, there was an average difference of −1.80 kg in the mean body weight change from baseline to 24 weeks between both group [70]. With linagliptin, the average changes in body fat mass over the course of 24 weeks were 0.36 ± 0.44 kg (*p* = 0.421) and the mean changes in body weight were 0.30 ± 0.21 kg (*p* = 0.173) compared with baseline, which were not statistically significant changes [70]. There was no statistically significant difference between the linagliptin and empagliflozin groups between baseline and 24 weeks in mean weight, visceral fat mass, or subcutaneous fat mass [70]. According to findings from various research, linagliptin either had a neutral weight effect or a little weight loss that did not statistically vary from groups taking other drugs.

##### Alogliptin

Oral DPP-4 inhibitor alogliptin (Figure 1g) has largely been studied in phase II/III trials with T2DM patients [71]. Whether used as a monotherapy or in combination with other anti-diabetic drugs, alogliptin has demonstrated a decrease in HbA1c and is generally well-tolerated [71]. Alogliptin has generally favorable safety characteristics, a low risk of hypoglycemia, and effects that are weight-neutral [71]. In patients with T2DM who were not properly controlled on stable-dose metformin, a multicenter, double-blind, active-controlled research examined the long-term effectiveness of alogliptin in comparison to glipizide in conjunction with metformin [72]. In the study, alogliptin 12.5 mg (*n* = 880), alogliptin 25 mg (*n* = 885), or glipizide 5 mg (*n* = 874) once daily, titrated to a maximum of 20 mg, were given to 2639 individuals at random for 2 years [72]. From week 26 through the completion of the trial, weights declined in the 2 alogliptin treatment groups by a range of 0.60 kg to 0.94 kg, but increased in the glipizide group during the same time period by a range of 0.86 kg to 0.97 kg [72]. The differences in weight change between glipizide and alogliptin dosages were all statistically significant [72]. In total, 84 Japanese people with poorly controlled T2DM were included in randomized research to compare the effectiveness of alogliptin (25 mg, once daily) and metformin (1000 mg, twice daily) on their body composition [73]. Alogliptin or metformin were given to participants in a 1:1 ratio [73]. Alogliptin increased body weight (from 66.5 ± 19.2 kg to 67.6 ± 19.3 kg), BMI (from 25.4 ± 6.1 kg/m^2^ to 25.8 ± 6.3 kg/m^2^), and fat mass (from 20.3 ± 12.8 kg to 21.8 ± 14.5 kg) substantially more than metformin did [73]. As stated previously, alogliptin increases weight while metformin decreases it in direct proportion to BMI [73]. On the other hand, patients with poorly controlled T2DM were randomly assigned to receive either glimepiride (*n* = 35), alogliptin monotherapy (*n* = 31), or alogliptin-pioglitazone (*n* = 33) therapy for 24 weeks in a 3-arm, multicenter, open-label, randomized, controlled trial [74]. The findings showed no discernible changes in body weight between the studied groups [74]. The efficiency of combining alogliptin with metformin and sulfonylurea with other DPP-4 inhibitors in 1887 T2DM patients was compared in a comprehensive review and meta-analysis [75]. According to the study’s findings, when alogliptin 25 mg was combined with metformin and sulfonylurea it was associated with the least mean weight gain compared to placebo (0.14 kg over 26 weeks) and compared with other DPP-4 inhibitors that was associated with a mean weight difference reduction of 0.60 kg, 0.80 kg, 0.33 kg and 1.10 kg for vildagliptin, saxagliptin, linagliptin, and sitagliptin, respectively [75]. More research is required to determine the precise impact of alogliptin on weight because the studies found inconsistent effects on body weight.

#### 3.1.4. SGLT-2 Inhibitors

The secondary active co-transporters that regulate renal sodium and glucose reabsorption are the sodium-dependent glucose cotransporter proteins-1 (SGLT1) and SGLT-2 [76]. Ninety percent of the reabsorption of glucose is carried out by SGLT-2, which are found in the early proximal renal tubule of the kidney [76]. The SGLT-1 transporter in the late proximal tubule reabsorbs the final 10% of the glucose [76]. SGLT-2 inhibitors are drugs that offer a means of managing hyperglycemia in T2DM patients using insulin-independent mechanism [77]. By blocking up to around 50% of the glucose reabsorption from the proximal tubule in the nephron, these therapeutic medicines increase urine glucose excretion and cause glucosuria (Figure 2) [77]. Even in people without diabetes, SGLT-2 inhibitors were shown to be effective in lowering body weight and had good effects on avoiding cardiovascular and renal illnesses [78,79]. Despite the fact that SGLT-2 inhibitors have been demonstrated to reduce body weight, this effect is relatively moderate due to compensatory mechanisms that work to keep the body weight constant [79]. As a result, researchers have hypothesized that SGLT-2 inhibitor co-administration with medicines that reduce food intake will be a more successful weight loss therapy than SGLT-2 inhibitor monotherapy [79].

##### Canagliflozin

Canagliflozin (Figure 1h) is SGLT 2 inhibitor that has received FDA approval to treat hyperglycemia in T2DM patients with diet and exercise [80]. Moreover, it has been demonstrated to lower systolic blood pressure (BP), HbA1c, and body weight [81]. A once-daily dose of 100 mg or 300 mg of canagliflozin or a placebo was randomly assigned to participants (*n* = 584) in a double-blind, placebo-controlled, phase 3 research [82]. Use of canagliflozin at doses of 100 mg and 300 mg resulted in considerable weight loss of 2.2% and 3.3% of the starting weight, respectively [82]. Another double-blind, randomized trial investigated the effects of glimepiride 6 mg or 8 mg daily for 104 weeks with canagliflozin 100 mg and 300 mg in 1450 T2DM patients [83]. With a reduction in baseline weight of 4.1% and 4.2% for canagliflozin doses of 100 mg and 300 mg, respectively, both canagliflozin doses significantly reduced body weight [83]. Whereas glimepiride was linked to a 0.9% increase in body weight [83]. A comprehensive systemic review and meta-analysis of 1904 studies examined the drug canagliflozin. In this study, 13,158 T2DM patients were examined, with 7859 being given canagliflozin and 5478 being placed in the control group [84]. They came to the conclusion that canagliflozin had demonstrated a significant reduction in body weight with both 100 mg and 300 mg when compared to the control (weighted mean difference: −3.32 kg). However, there was no dose dependence for this effect [84]. In total, 1.2 mg of liraglutide, 100 mg of canagliflozin, or a combination of liraglutide and canagliflozin were given to a total of 45 T2DM patients using metformin with or without sulfonylurea in 2020 for 16 weeks [85]. Each of the three groups demonstrated a discernible weight loss impact [85]. The combination of liraglutide 1.2 mg with canagliflozin 100 mg showed an additive effect on weight loss, but not on HbA1c [85]. According to the study’s findings, subjects receiving liraglutide, canagliflozin, or the combination of liraglutide and canagliflozin, showed a body weight reduction of 1.9 ± 0.8 kg, 3.5 ± 0.5 kg, and 6.0 ± 0.8 kg, respectively [85]. Hence, it is believed that canagliflozin has a fairly favorable impact on weight loss.

##### Dapagliflozin

Dapagliflozin (Figure 1i) selectively and reversibly inhibits the SGLT-2 transporter [86]. The FDA has approved dapagliflozin for the treatment of hyperglycemia in adult T2DM patients as an adjunct to diet and exercise [87]. By enhancing renal glucose excretion and resulting in calorie loss, dapagliflozin has had a beneficial effect on glycamic management, systolic BP, and weight loss [88]. Dapagliflozin 10 mg daily was compared to a placebo and added to open-label metformin for approximately 6 months in a multicenter, multinational, randomized trial with 182 T2DM patients [89]. Over the course of the study, dapagliflozin 10 mg daily added to metformin resulted in significantly greater weight loss (−2.96 kg in the dapagliflozin 10 mg group vs. −0.88 kg in the placebo group) and better glycemic control than placebo added to metformin [89]. Whole body fat loss and a decrease in the volumes of visceral adipose tissue and subcutaneous adipose tissue were found to be responsible for the effect on body weight [89]. In another double-blind, randomized research, 50 obese adults with prediabetes were compared to placebo for 24 weeks to see how oral dapagliflozin 10 mg once daily and subcutaneous long-acting exenatide 2 mg once weekly affected their body weight [90]. In comparison to the placebo group, the average weight was significantly lowered by the combined therapy of dapagliflozin/exenatide (4.13 kg after 24 weeks), and it was well tolerated [90]. In 17,160 T2DM patients who were monitored for a median of 4.2 years, dapagliflozin was compared to a placebo in a phase 3 double-blind, international, randomized study named DECLARE-TIMI 58 [91]. When compared to a placebo, dapagliflozin significantly reduced cardiovascular death, heart failure hospitalizations, and body weight, with a difference of 1.8 kg [91]. Using 100 T2DM patients on metformin monotherapy, a recent randomized, double-blind trial examined the impact of dapagliflozin 10 mg daily on epicardial fat thickness as a primary result and body weight as a secondary outcome [92]. The participants were randomized to receiving metformin up to 1000 mg twice daily or receiving dapagliflozin 10 mg/day [92]. The thickness of the epicardial fat was immediately and considerably reduced by dapagliflozin, which also caused a 3.5 kg weight loss in the body. However, it is possible that weight loss is not necessary for the decrease in epicardial fat thickness [92]. Generally, dapagliflozin has modest effects on weight loss.

##### Empagliflozin

One inhibitor of the SGLT-2 transporter is empagliflozin (Figure 1j) [93]. In August 2014, it gained FDA approval to enhance glycemic management add on to diet and exercise [93]. In order to assess the impact of empagliflozin on body weight and adiposity in individuals with T2DM, a study evaluated two cohorts of patients from five randomized trials [94]. There were 3300 patients in total between the 2 cohorts [94]. In 1 of these studies, 823 T2DM patients (cohort 1) were randomly assigned to receive either empagliflozin 10 mg, empagliflozin 25 mg, or a placebo every day for 12 weeks [95]. In total, 2477 T2DM patients from 4 studies made up cohort 2 [96,97,98,99]. In these studies, patients were randomly assigned to receive either monotherapy (empagliflozin 10 mg/day, empagliflozin 25 mg/day, or placebo/day) [96], or add-on medications to metformin, metformin with sulfonylurea, or pioglitazone with or without metformin for 24 weeks [97,98,99]. This study’s findings showed that empagliflozin significantly decreased weight, waist circumference, and adiposity in both cohorts of T2DM patients as compared to placebo [94]. The placebo-adjusted average change in body weight with empagliflozin was −1.7 kg in cohort 1 and −1.9 kg in cohort 2 [94]. In a recent randomized trial, empagliflozin 10 and 25 mg were combined to metformin and tested on 637 adult patients with T2DM [100]. Empagliflozin 10 mg (*n* = 217), empagliflozin 25 mg (*n* = 213), or a placebo (*n* = 207) were given to the participants [100]. According to the trial’s findings, both dosages of empagliflozin significantly lowered HbA1C, body weight, and systolic BP over the course of both the short-term (24 weeks) and long-term (76 weeks) [100]. Body weight loss was strongly correlated with patients’ initial weight [100]. They came to the conclusion that empagliflozin can help people lose about 4–5 kg of weight over time [100]. A total of 545 subjects with T2DM participated in a recent observational, retrospective, cohort trial that assessed the effects of empagliflozin and liraglutide on weight loss outcomes [101]. With an average weight loss of 2.81 kg in the empagliflozin group (*n* = 247) and 2.17 kg in the liraglutide group (*n* = 298) after one year [101], both the empagliflozin (*n* = 247) and liraglutide (*n* = 298) groups have demonstrated a weight loss impact. However, there was no discernible difference between the groups treated with empagliflozin and liraglutide after a year [101]. Thus, empagliflozin has demonstrated clinically significant effects on weight loss.

##### Ertugliflozin

The newest SGLT-2 transporter inhibitor FDA-approved for the treatment of T2DM is ertugliflozin (Figure 1k), which is thought to be similarly safe to other SGLT-2 inhibitors [102]. Ertugliflozin has been linked to weight loss in several studies [103,104,105]. For instance, ertugliflozin’s ability to cause weight loss in 461 individuals with T2DM was examined as a secondary endpoint in a phase 3, double-blind, multicenter, randomized trial [104]. Ertugliflozin 5 mg, ertugliflozin 15 mg, or a placebo were given to participants at random (1:1:1) once daily [104]. After 26 weeks of treatment, both ertugliflozin doses significantly decreased body weight as compared to the placebo [104]. The placebo-adjusted average body weight differences from baseline were −1.76 kg for ertugliflozin 5 mg and −2.16 kg for ertugliflozin 15 mg [104]. Ertugliflozin’s impact on body weight after 26 weeks was examined in another randomized, phase 3, double-blind, multicenter trial with 621 T2DM individuals [103]. As a supplement to metformin, participants had been randomly assigned (1:1:1) to receive either ertugliflozin 5 mg once day, ertugliflozin 10 mg once daily, or a placebo [103]. The mean body weight changes from baseline were −1.3 kg for placebo, −3 kg for ertugliflozin 5 mg, and −2.9 kg for ertugliflozin 15 mg [103]. When combined with metformin, ertugliflozin considerably reduced weight when compared to placebo, and both doses had a similar, significant impact on body weight [103]. The phase 3 VERTIS SITA2 trial randomly assigned 464 T2DM patients managed with metformin and sitagliptin to receive ertugliflozin 5 mg, 15 mg, or placebo [105]. The addition of ertugliflozin to metformin and sitagliptin in this trial resulted in a significant decrease in participant body weight, with an average weight loss of 3.4 kg in the ertugliflozin group compared to 1.3 kg in the placebo group [105]. In a randomized research that lasted 52 weeks and included T2DM patients being treated with metformin, the effects of co-administration of ertugliflozin and sitagliptin compared to the individual medicines had been assessed [106]. A total of 1233 individuals were randomly assigned to receive sitagliptin 100 mg, ertugliflozin 5 mg, ertugliflozin 15 mg, or a combination of ertugliflozin 5 mg and sitagliptin 100 mg, or ertugliflozin 15 mg with sitagliptin 100 mg [106]. Ertugliflozin 5 mg/sitagliptin 100 mg and ertugliflozin 15 mg/sitagliptin 100 mg were compared to only sitagliptin 100 mg alone for the body weight endpoint [106]. When compared to sitagliptin 100 mg alone, ertugliflozin 5 mg/sitagliptin 100, and ertugliflozin 15 mg/sitagliptin 100 both reduced body weight clinically meaningfully [106]. The average initial weight loss for these 2 dosages was 3% and 4.2%, respectively. Moreover, ertugliflozin’s effect on body weight persisted through week 52 [106]. As a result, ertugliflozin has significant weight loss effects.

#### 3.1.5. GLP-1 Receptor Agonists

GLP-1 receptor agonists resemble endogenous GLP-1 (Figure 2), but with longer half-lives [107]; they boost-cell proliferation, slowing stomach emptying, secrete more insulin, and suppress appetite [107]. As a result, they enhance glycemic management and aid in weight loss [107]. Exendin-4, a 39 amino acid with 53% similarity to human GLP-1, was isolated from the glia monster, which sparked the discovery of GLP-1 receptor agonists [107,108]. A synthetic peptide of Exendin-4, called exenatide, was introduced as the first GLP-1 in 2005 under the trade name Byetta^®^ because of its high degree of stability [107,108]. The most often reported side effects of these therapeutic drugs that force patients to stop receiving treatment are gastrointestinal [109,110]. In this analysis, we emphasize the six GLP-1 receptor agonists’ remarkable weight loss outcomes across a range of clinical research programs.

##### Exenatide

Exenatide, a synthetic peptide identical to exendin-4, was the first GLP-1 agonist receptor to receive FDA approval [107,108,111]. The homology to human GLP is 53% [111]. It was authorized to assist T2DM patients in achieving better glycemic control [111]. It was initially given as a 10 µg twice-daily subcutaneous injection under the brand name Byetta [111,112]. However, a long-acting new formulation using the 2 mg injectable microsphere technology was authorized in 2012 under the brand name Bydureon™ [111]. For the two doses, the glycemic control was remarkably identical. However, there were fewer gastrointestinal side effects and higher patient compliance with the long-acting formulation [113].

This review describes exenatide’s effectiveness in reducing weight in people with uncontrolled T2DM who participated in the Diabetes therapy Utilization: Researching changes in A1C, weight and other factors (DURATION) clinical trials [114]. In DURATION-1 [114], once-weekly exenatide and twice-daily exenatide were compared [114]. In this 30-week, multicenter, open-labeled, comparator-controlled phase 3 trial, 295 patients with uncontrolled type 2 diabetes who were taking oral medications or were drug-naïve were randomly assigned to receive either 2 mg of subcutaneous exenatide once a week or 10 µg of subcutaneous exenatide twice a day [114]. For the once-weekly and twice-daily doses of exenatide, the average weight loss at week 30 was 3.7 kg and 3.6 kg, respectively. The daily dose, however, was associated with more gastrointestinal adverse effects [114]. In DURATION-2, a 26-week, randomized, double-blinded, parallel-group trial, 491 T2DM patients on metformin were randomized at 1:1:1 to receive 2 mg of once-weekly subcutaneous exenatide, sitagliptin, or pioglitazone [115]. The efficacy and safety of once-weekly exenatide were compared to those of sitagliptin and pioglitazone. For the exenatide and sitagliptin groups, the average weight loss was 2.3 kg and 0.8 kg, respectively. As opposed to a gain of 2.8 kg on average in the pioglitazone group [115]. In DURATION-3, a 26-week, randomized, open-labeled, parallel research, 456 T2DM patients using metformin with/without sulfonylurea were compared to once-weekly exenatide and insulin glargine [116]. Exenatide 2 mg once weekly or insulin glargine were given to patients in a 1:1 randomization scheme [116]. In comparison to the glargine group, which experienced an average gain of 1.4 kg, the exenatide group experienced an average weight reduction of 2.6 kg [116]. In the 26-week, randomized, double-blind, phase 3 trial DURATION-4, the safety and effectiveness of once-weekly exenatide were compared to those of metformin, sitagliptin, and pioglitazone in drug-naïve T2DM patients [117]. A total of 820 patients were randomly assigned to receive either 2 mg of subcutaneous exenatide once a week and an oral placebo (*n* = 248), or 2000 mg of metformin every day and a subcutaneous placebo (*n* = 246), or 45 mg of pioglitazone every day and a subcutaneous placebo (*n* = 163), or 100 mg of sitagliptin every day and a subcutaneous placebo (*n* = 163) [117]. At 26 weeks, the exenatide and sitagliptin groups had average weight losses of 2 kg and 0.8 kg, respectively. While the pioglitazone group experienced an average gain of 1.5 kg [117]. The DURATION-5 trial, an open-label, randomized, multicenter, parallel-Group trial, had a similar goal to the DURATION-1 trial: to compare the efficacy and safety of 2 mg of exenatide once per week to that of 10 µg of exenatide twice daily [118]. However, the primary outcomes were evaluated at 24 weeks rather than 30 weeks [118]. In total, 252 T2DM patients were randomly assigned to receive either 10 µg subcutaneous exenatide twice daily or 2 mg exenatide once per week [118]. Weekly exenatide was linked to better glycemic control and fewer adverse effects after 24 weeks [118]. Both led to weight loss, with the twice-daily and once weekly exenatide causing weight loss of an average of 1.4 kg and 2.3 kg, respectively [118]. The DURATION-6 trial, a 26-week, randomized, open-label, parallel-group experiment in which 912 T2DM patients on one or more oral medications were randomized at 1:1 to receive either 2 mg of exenatide once weekly or 1.8 mg of liraglutide once daily, compared exenatide to liraglutide [119]. The liraglutide group has demonstrated greater results on weight loss [119]. For the exenatide and liraglutide groups, the average weight loss was 2.68 kg and 3.57 kg, respectively [119]. In DURATION-7, a randomized, double-blind, multicenter, placebo-controlled, parallel group, phase 3 trial [120], exenatide was introduced to insulin-treated T2DM patients once weekly. A 2:1 randomization of 2 mg of exenatide once weekly or a placebo was performed in 464 uncontrolled T2DM patients on insulin glargine [120]. The exenatide group had better glycemic control and had lost an average of 1.5 kg more weight than the placebo group, according to the data [120]. In the long-term, randomized, open-label, parallel-group, comparator-controlled, phase 3 trial DURATION-Neoadjuvant-1 (DURATION-NEO-1), the effectiveness and tolerability of the new formulation of the long-acting exenatide were investigated [121]. A total of 375 T2DM patients who were taking oral medications or were drug-naïve were randomly assigned to receive either the novel formulation of exenatide once per week or twice daily for 28 weeks [121]. The weekly version was inferior to the glycemic control profile [121]. No discernible difference was seen between the two groups’ average weight loss, which was 1.49 kg for the once-weekly exenatide group and 1.89 kg for the twice-daily exenatide arm [121]. In DURATION-NEO-2, an open-label, randomized, multicenter trial [122], the novel formulation of exenatide was compared to sitagliptin and a placebo for effectiveness, safety, and tolerability. Exenatide 2 mg once weekly, sitagliptin 100 mg, or placebo were randomly administered to 365 patients with uncontrolled T2DM who were taking metformin [122]. Exenatide outperformed the other groups in terms of glycemic control [122]. In comparison to the placebo group, which experienced an average gain of 0.15 kg, the exenatide and sitagliptin groups experienced average weight reductions of 1.12 kg and 1.19 kg, respectively [122].

A meta-analysis of six randomized clinical trials of exenatide effects on weight loss was conducted on a total number of 362 obese non-diabetic participants [123]. It was shown that exenatide was superior to control groups in weight loss with a mean difference of 4.47 kg (*p* < 0.0001) [123]. Moreover, greater BMI and waist circumference reductions were also reported in the exenatide groups with a mean difference of 0.86 kg/m^2^ (*p* = 0.001) and 1.78 cm (*p* = 0.009) for the BMI and waist circumferences, respectively [123]. After 12 weeks, patients who lost ≥ 5% of their weight were classified as high responders while who had lost ≥ 10% were classified as super-responders, and those who failed to achieve a 5% loss of their weight were classified as low responders [124]. The average weight loss for the exenatide group was 6.2 kg and 1.9 for high and low responders, respectively. While average weight loss for the hypocaloric diet was 7.2 kg and +0.6 kg for high and low responders, respectively [124]. Overall, exenatide has mild to moderate weight loss effect. The only significant difference between the daily and once-weekly doses was that the long-acting version had fewer side effects.

##### Liraglutide

With 97% amino acid sequence identical to human GLP-1, liraglutide was the second GLP-1 receptor agonist to be licensed [125,126]. In 2010, the FDA gave it the go-ahead for the treatment of T2DM [125]. The FDA approved a new dose of liraglutide 3 mg in 2014 for the treatment of obesity in patients without diabetes under the brand name Saxenda^®^ [125]. This approval was made possible by the impressive results of weight loss brought on by 3 mg liraglutide in clinical trials in T2DM patients (an average of 5% weight loss) [125,126,127]. Daily subcutaneous injections of liraglutide are given [127]. The most frequent adverse effects are gastrointestinal, which often go away over time [127]. Liraglutide has been the subject of numerous clinical trials; the first of these was the Liraglutide Effect and Action in Diabetes (LEAD) program for diabetics, which enrolled more than 4000 T2DM patients from 40 different nations [128,129]. The program’s goal was to evaluate the safety and effectiveness of liraglutide in T2DM patients when used alone or in combination with other antidiabetic medications [128,129]. Significant weight loss had been seen with liraglutide [128]. According to dual-energy X-ray absorptiometry and computed tomography results from a sub-study of LEAD 2, the bulk of weight loss originates from adipose tissue [130]. In LEAD 3; A double-blind, randomized phase 3 trial, 746 T2DM patients were randomly assigned to receive 1.2 mg of liraglutide (*n* = 251), 1.8 mg of liraglutide (*n* = 247), or 8 mg of glimepiride (*n* = 248) [131]. Liraglutide as a monotherapy for T2DM was the subject of this study, which sought to assess its efficacy and safety [131]. The liraglutide group experienced an average weight loss of 2 kg, whereas the glimepiride group experienced an average weight gain of 1 kg [131].

Liraglutide was observed to cause weight reduction that was proportionate to dose [132]. For instance, 533 T2DM patients taking 1000 mg of metformin and 4 mg of rosiglitazone twice daily were randomized at a 1:1:1 ratio to once daily doses of 1.2 mg or 1.8 mg of liraglutide or placebo in the 26-week, double-blind, placebo-controlled, parallel-group experiment known as LEAD 4 [132]. Liraglutide dosages of 1.2 mg and 1.8 mg resulted in average weight loss of 1 kg and 2 kg, respectively, while the placebo group experienced an average gain of 0.6 kg [132]. In total, 581 T2DM patients were randomly assigned to receive either 1.8 mg of liraglutide, a placebo, or insulin glargine in a 2:1:2 ratio as part of the 26-week LEAD 5 research, a randomized phase 3 study [133]. The liraglutide and placebo groups’ average weight loss was 1.80 kg and 0.42 kg, respectively, while the glargine group’s average weight gain was 1.60 kg [133]. Moreover, liraglutide and sitagliptin were compared in a 26-week open-label, parallel-group trial for individuals with uncontrolled T2DM [134]. In total, 665 patients were randomly assigned to receive either 100 mg of oral sitagliptin once daily, 1.2 mg, or 1.8 mg of liraglutide once daily [134]. With 1.2 mg, 1.8 mg, and 100 mg of liraglutide and sitagliptin, respectively, weight reductions of 2.86 kg, 3.38 kg, and 0.96 kg were reported [134]. In LEAD 6, a 26-week open-label phase 3 trial, 467 T2DM patients using metformin or a sulfonylurea were randomly assigned to receive either 1.8 mg of liraglutide once a day (*n* = 233) or 10 µg of exenatide twice a day (*n* = 231) [135]. With liraglutide and exenatide, the average weight loss was 3.24 kg and 2.87 kg, respectively [135]. In a 26-week, double-blind research, liraglutide was also assessed as an add-on therapy with insulin glargine with or without metformin [136]. A 1:1 randomization was used to assign 451 participants to receive either 1.2 mg, 1.8 mg liraglutide, or a placebo [136]. In comparison to the placebo group, the liraglutide group saw a weight loss of 3.5 kg on average [136]. In a global, double-blind research, 202 T2DM patients stabilized on SGLT-2 inhibitor received 1.8 mg of liraglutide as an add-on medication and were compared to the placebo group (*n* = 101) [137]. Liraglutide did not statistically differ from the placebo in terms of weight loss (2.81 kg in the liraglutide group versus 1.99 kg in the placebo group) [137]. The SCALE clinical studies on liraglutide were done in order to demonstrate its anti-obesity impact [138,139]. A randomized, double-blind, international, phase 3 trial is called SCALE DIABETES [138]. With a 2:1:1 ratio of liraglutide 3.0 mg, 1.8 mg, and placebo, a total of 846 patients were enrolled [138]. Only 21.4% of those who got placebo saw weight loss of 5% or more, compared to 54.3% of those who received 3mg of liraglutide and 40.4% of those who received 1.8 mg of liraglutide [138]. With 3 mg, 1.8 mg, and placebo, respectively, an average decrease of 6.4 kg, 5 kg, and 2.2 kg was seen at week 56 [138]. In total, 396 individuals were randomly assigned to receive 3 mg of liraglutide or a placebo at a 1:1 ratio in a global, 56-week-long phase 3 experiment termed the SCALE INSULIN trial [139]. At 56 weeks, liraglutide 3.0 mg had a mean weight reduction of 5.8% compared to placebo’s 1.5%, and 51.8% of patients in the liraglutide group lost 5% of their body weight as opposed to 24.0% in the placebo group [139]. A critical review assessed the effectiveness and safety of liraglutide for weight loss in non-diabetic patients [140]. Their analysis comprised a total of 5 randomized, placebo-controlled studies [140,141]. When used in conjunction with lifestyle changes, liraglutide caused weight loss of 4–6 kg on average [140,141]. Although it was less effective than the phentermine/topiramate combination, liraglutide showed superior weight loss results versus orlistat and lorcaserin [140,141]. However, due to gastrointestinal side effects, it had significant withdrawal rates [140,141]. Liraglutide has an advantage over other anti-obesity medications in that it produces superior glycemic results [140,141]. In addition, five randomized clinical studies with 1758 people randomly assigned to placebo and 2996 participants receiving liraglutide were included in a systemic review and meta-analysis [141]. A statistically significant average weight loss of 5.52 kg with liraglutide in overweight persons was found to be effective and safe [141].

Another meta-analysis that looked at 31 studies involving 8060 overweight adults revealed that liraglutide had a weight-loss impact, with an average difference of 4.19 kg between liraglutide and placebo in body weight loss [142]. The mean differences in waist circumference and BMI were, respectively, 3.11 cm and 1.55 kg/m^2^. Indicating cardioprotective qualities, liraglutide also decreased low-density lipoprotein cholesterol and diastolic blood pressure [142]. We came to the conclusion that liraglutide significantly reduced body weight, which resulted in the FDA approving the 3 mg dose as an anti-obesity medication. Nonetheless, gastrointestinal side effects continue to be the main reason for discontinuation.

##### Dulaglutide

Under the trade name Trulicity^®^, dulaglutide, a GLP-1 receptor agonist, was authorized by the FDA in 2014 [143]. It is composed of a DPP-IV-protected GLP-1 analogue covalently attached to a human IgG4-Fc heavy chain via a small peptide linker [143]. A 1.5 mg subcutaneous injection is given once each week [143]. As an adjunctive therapy to other oral antihyperglycemic medications or/and insulins, clinical trials have demonstrated its efficacy and typically good tolerability [144]. In high-risk individuals, such as those with obesity-related cardiovascular and chronic renal illnesses, it exhibits positive outcomes [144,145]. Similar to other GLP-1 agonists, adverse effects are typically gastrointestinal [144,146].

We present the effectiveness of dulaglutide in weight loss from the AWARD clinical trials on uncontrolled T2DM in this review [147,148]. In AWARD-1, a multicenter, parallel-arm, phase 3 clinical research, 978 patients stabilized on metformin and pioglitazone were randomly assigned at 2:2:2:1 into 4 arms: 1.5 mg or 0.75 mg dulaglutide, 10 µg exenatide, or placebo, [147]. Exenatide and 1.5 mg dulaglutide both caused weight loss at 26 weeks, with average losses of 1.07 kg and 1.3 kg, respectively [147]. However, patients on 0.75 mg dulaglutide and placebo gained an average weight of 0.20 kg and 1.24 kg, respectively [147]. In AWARD-2, an open-label clinical trial, 810 metformin-taking patients with type 2 diabetes were randomly assigned at a ratio 1:1:1 to receive either 1.5 mg, 0.75 mg of dulaglutide once a week or daily insulin glargine [148]. In comparison to insulin glargine, dulaglutide induced an average weight loss of 1.87 kg and 1.33 kg for the 1.5 mg and 0.75 mg doses, respectively, after 52 weeks [148]. Dulaglutide and metformin were compared in AWARD-3, a double-blind, randomized clinical trial on 807 T2DM patients receiving just one oral antidiabetic medication [149]. In a 1:1:1 ratio, patients were randomly assigned into three groups: metformin; 1.5 mg dulaglutide; or 0.75 mg dulaglutide [149]. Weight loss at week 26 was 1.36 kg for the 0.75 mg dose of dulaglutide, compared to 2.29 kg for the high dose of dulaglutide and 2.22 kg for the high dose of metformin [149]. Moreover, AWARD-4, a phase 3 randomized, double-blind trial in which all participants were taking prandial insulin lispro, was conducted on T2DM patients. In total, 884 patients in total were randomly assigned to receive daily doses of 1.5 mg, 0.75 mg dulaglutide once per week, or daily doses of insulin glargine at bedtime [150]. With dulaglutide 1.5 mg, the average weight was lowered by roughly 0.9 kg compared to a 2.3 kg rise with glargine [150]. A weight recovery was seen in the dulaglutide 0.75 mg group after an initial weight loss in the group. The rise, however, was considerably less than in the glargine group [150]. In the AWARD-5, a randomized, double-blind, multicenter, parallel-arm clinical trial in which 1098 metformin-treated T2DM subjects were randomly assigned to two doses of dulaglutide, sitagliptin, or placebo [151], sitagliptin and dulaglutide were contrasted Average weight losses of 2.88 kg, 2.39 kg, and 1.75 kg were seen after 104 weeks for 1.5 mg, 0.75 mg dulaglutide, and sitagliptin, respectively [151].

A double-blind, randomized, parallel-arm clinical trial called AWARD-6 compared dulaglutide to another GLP-1 agonist, liraglutide [152]. In total, 599 T2DM patients using metformin were randomly assigned to receive either 1.8 mg per day of liraglutide or 1.5 mg per week of dulaglutide [152]. When compared to dulaglutide, liraglutide produced better weight loss results, with an average loss of 3.61 kg as opposed to 2.90 kg in the dulaglutide group [152]. Dulaglutide’s renal protective properties were investigated in AWARD-7, because they are not dependent on renal clearance [146]. In a multicenter, open-label clinical research, 577 T2DM patients with moderate to severe chronic renal disease were randomly assigned to receive either insulin or insulin plus an oral drug in a 1:1:1 ratio of 0.75 mg, 1.5 mg dulaglutide once weekly, and insulin glargine [146]. The dulaglutide group showed more kidney-protective outcomes. Those receiving glargine experienced weight gain, while those using dulaglutide experienced weight decrease [146]. Despite the fact that the mean weight in the dulaglutide group decreased considerably from baseline in AWARD-8, no clinically significant weight loss was seen when 239 T2DM patients taking glimepiride were compared to a placebo group [153]. According to the AWARD-9 clinical study, 1.5 mg of dulaglutide caused a 1.91 kg loss in 300 T2DM patients receiving insulin glargine with or without metformin, while individuals on a placebo gained an average of 0.50 kg [154]. In AWARD-10, a global, double-blind, randomized phase 3 trial, 424 T2DM patients receiving SGLT-2 inhibitors with or without metformin were randomized into groups receiving 1.5 mg, 0.75 mg dulaglutide, or placebo in a 1:1:1 ratio [155]. For doses of 1.5 mg and 0.75 mg dulaglutide and placebo, the weight decreases after 24 weeks were 3.1 kg, 2.6 kg, and 2.1 kg, respectively [155]. A randomized, double-blind, parallel-arm phase 3 clinical trial called AWARD-11 was presented to examine the effectiveness and safety of large dosages of dulaglutide [156]. Patients on metformin for uncontrolled T2DM made up the trial population [156]. Dulaglutide dosages of 4.5 mg, 3 mg, or 1.5 mg were given once weekly for 52 weeks to 1842 patients [156]. The dose of dulaglutide was inversely correlated with changes in HbA1c and weight [156]. When compared to patients who were kept on 1.5 mg of dulaglutide, patients who utilized higher dosages of the drug experienced greater weight reduction effects [156]. With the dosages of 3 mg and 4.5 mg dulaglutide, the weight loss was 0.9 kg (*p* = 0.001) and 1.6 kg (*p* < 0.001), respectively [156]. For all doses, the safety profile and side effects were essentially the same [156]. Dulaglutide was tested in children aged 10 to 17 years old treated with metformin or insulin in a clinical trial known as AWARD-PEDS as T2DM rates are rising in the young population [157]. A 26-week, double-blind, placebo-controlled trial had 154 participants [157]. The patients were randomly assigned to receive 0.75 mg, 1.5 mg dulaglutide, or placebo [157]. The two dulaglutide-treated groups’ glycated hemoglobin levels were found to be lower [157]. BMI, however, showed no difference [157]. In conclusion, dulaglutide had demonstrated a considerable reduction in weight that was proportional to the dose employed. The weight loss benefits of the 1.5 mg dulaglutide were superior to those of exenatide and were essentially identical to those of metformin. However, it was not as good as liraglutide. The most frequent side effects were gastrointestinal.

##### Lixisenatide

A once-daily prandial GLP-1 agonist, lixisenatide is marketed under the trade name Lyxumia^®^ and received approval in the European Union, Mexico, Japan, and other nations in 2013 [158]. It consists of 44 amino acids and is based on the exendin-4 structure with some modifications [158]. Eventually, it was granted approval by the European Medicines Agency (EMA) in 2016 under the trade name Adlyxine^TM^ [159]. In T2DM patients, it is utilized as an adjunct to lifestyle therapies. Moreover, in the case of uncontrolled T2DM, it can be used with additional oral medications or basal insulin therapy [158]. A 20 µg subcutaneous injection once daily maintenance dose of lixisenatide is administered after two weeks of gradually increase of the dose [160]. The most frequent adverse effects, nausea and vomiting, usually subside with time [160]. Low incidence of hypoglycemia is also linked to lixisenatide [161].

We summarize the lixisenatide weight reduction results from the GetGoal clinical trials in this review. In GetGoal-Mono, a 12-week, randomized, double-blinded trial in which 361 participants were randomized into a 2-step titration, lixisenatide was compared to a placebo in drug-naïve T2DM patients [162]. Lixisenatide was given in the following dosages: 10 μg for 1 week, 15 μg for 1 week, and then 20 μg (for *n* = 120), or one-step titration; 10 μg for 2 weeks and then 20 μg (for *n* = 119); or a placebo group (for *n* = 122) [162]. In all groups, weight losses were on average 2 kg. In the lixisenatide group, nausea was more prevalent [162]. In the GetGoal-M study, which was a 24-week, double-blinded, randomized, placebo-controlled phase 3 trial [163], lixisenatide was examined as an addition to metformin in individuals with uncontrolled T2DM. In total, 680 participants in all were randomly assigned to receive 20 μg of lixisenatide in the morning, evening, or a placebo [163]. Between the lixisenatide and placebo groups, there was a discernible difference in glycemic control [163]. Nonetheless, the average weight loss across all groups was 2 kg, with no statistically significant difference [163]. In GetGoal-X, a 24-week, randomized, open-label research, 634 metformin-using T2DM patients were randomly assigned to receive either 20 μg of lixisenatide once daily or 10 μg of exenatide twice daily [164]. With lixisenatide and exenatide, the average weight decreases were 2.96 kg and 3.98 kg, respectively [164]. The exenatide group had improved glycemic control. However, lixisenatide group members experienced less hypoglycemia and gastrointestinal adverse effects (*p* < 0.05) [164]. GetGoal-F1, a randomized, parallel-group, double-blinded, multicenter, phase 3 trial, evaluated the impact of 2 titration regimens on the tolerance of lixisenatide [165]. In total, 484 T2DM patients using metformin were randomized to receive lixisenatide in a one-step dose titration, two-step dose titration, or placebo [165]. Average weight loss was 2.6 kg, 2.7 kg, and 1.6 kg for the one-step lixisenatide group, two-step lixisenatide, and the combined placebo group, respectively [165].

GetGoal-S clarified the effectiveness and safety of lixisenatide in T2DM patients on metformin and a sulfonylurea [166]. In this 24-week, parallel-group, double-blinded experiment, 855 T2DM patients were randomly assigned to receive either placebo or 20 µg of lixisenatide subcutaneously once daily (following the 2-step titration method) [166]. In the lixisenatide group, there was an average weight loss of 1.76 kg compared to 0.93 kg in the placebo group [166]. Moreover, GetGoal-P clarified the effectiveness and safety of lixisenatide in T2DM patients using metformin and pioglitazone [167]. In total, 484 individuals were randomly assigned to receive either a placebo or 20 μg of lixisenatide once a day in this multicenter, randomized, double-blinded research [167]. The lixisenatide group had superior glycemic control [167]. However, the average weight reduction was 0.2 kg in lixisenatide group and 0.2 kg gain in the placebo group without any significant differences [167]. In GetGoal-L, a global, double-blinded, randomized, phase 3-trial, 495 T2DM patients on basal insulin therapy were randomly assigned to receive either placebo or 20 μg of lixisenatide once day (following the 2-step titration approach) [168]. For lixisenatide and the placebo, the average weight loss was 1.8 kg and 0.5 kg, respectively [168]. The lixisenatide group experienced gastrointestinal side effects and a few instances of hypoglycemia (4 from 328) [168]. In GetGoal-Duo 1, a global, randomized, double-blind, parallel-group, placebo-controlled, phase 3 trial where HbA1c levels are still high despite the addition of insulin glargine upon 12 weeks, the effectiveness of lixisenatide in uncontrolled T2DM patients on oral medicines was studied [169]. The 2-step titration approach was used to randomly assign 446 T2DM patients to either placebo or 20 μg of lixisenatide once daily [169]. With superior glycemic control than a placebo (*p* < 0.0001), lixisenatide has the potential to replace prandial insulin [169]. A statistically significant difference (*p* = 0.0012) between the average weight gain of 1.2 kg in the placebo group and a weight gain of 0.3 kg in the lixisenatide group [169]. However, the therapy group experienced greater gastrointestinal side effects and hypoglycemia [169]. In GetGoal-Duo 2, the effectiveness of lixisenatide was compared to prandial insulin administered once or three times each day [170]. A 26-week, open-label, three-parallel-arm, randomized trial in which 298 uncontrolled T2DM patients on oral medications and insulin glargine are randomly assigned at a ratio of 1 to 1 to 20 mg of lixisenatide or prandial insulin once, twice, or three times daily [170]. The lixisenatide group saw an average weight loss of 0.6 kg, whereas the insulin glulisine once-daily and three-times-daily groups experienced average weight gains of 1 kg and 1.4 kg, respectively [170]. GetGoal-O, a global, randomized, placebo-controlled, double-blind, phase 3 trial in which 350 patients were allocated at random to receive lixisenatide or a placebo, evaluated the efficacy and safety of lixisenatide in T2DM patients aged 70 years or older [171]. For the lixisenatide group and placebo, the average weight loss was 1.47 kg and 0.16 kg, respectively [171]. In a randomized, double-blind, parallel-group trial, 319 obese T2DM patients under 50 years old on metformin were randomly assigned at 1:1 to receive either 20 µg of lixisenatide once daily or 100 mg of sitagliptin once daily [172]. With lixisenatide and sitagliptin, the average weight loss at 24 weeks was 2.5 kg and 1.2 kg, respectively [172]. A 12-week, multicenter, randomized, open-labeled, controlled trial with T2DM patients compared lixisenatide and basal insulin to numerous insulin injections [173]. A 1:1 allocation of lixisenatide, insulin glargine, and multiple daily insulin was given to 31 Japanese patients with T2DM [173]. In comparison to the multiple daily insulin group, which experienced an average gain of 0.8 kg, the lixisenatide/basal insulin combination group experienced a mean body weight reduction from baseline of 2.5 kg [173]. In a 10-week, randomized, parallel-group, investigator-blinded experiment, the effects of Lixisenatide and liraglutide were contrasted with regard to macronutrient intake, gastrointestinal side effects, and pancreatic function [174]. In total, 50 T2DM patients with a BMI of 18 to 40 kg/m^2^ were randomly assigned to receive either 1.8 mg of liraglutide or 20 µg of lixisenatide once daily [174]. Both therapies demonstrated better pancreatic function, decreased food intake, and decreased weight after 10 weeks. However, the average weight loss in the liraglutide group (3.6 kg) was considerably greater than in the lixisenatide group (1.9 kg) [174]. Overall, the GetGoal data showed that lixisenatide had significantly reduced weight loss in the majority of studies. However, when compared to exenatide and liraglutide, it did provide a smaller weight loss.

##### Semaglutide

A GLP-1 receptor agonist, semaglutide shares 94% of its amino acid sequence with human GLP-1 [175]. Semaglutide’s half-life is between 155 and 184 h, allowing for once-weekly dosing [176]. Under the trade name Ozempic^®^, subcutaneous semaglutide once per week was approved by the FDA in 2017 for T2DM [177]. Under the trade name Rybelsus^®^ [178], oral semaglutide for T2DM was approved by the FDA in 2019 with a maximum dose of 14 mg [178]. The FDA authorized Wegovy^®^, an anti-obesity drug, in 2021 as a supplement to lifestyle therapies [179]. It contains 2.4 mg subcutaneous semaglutide once per week [179].

The safety and effectiveness of semaglutide have been evaluated in a number of programs and clinical trials [180,181,182,183,184,185,186,187,188]. The SUSTAIN program is the first one [180,181,182,183,184,185,186]. Efficacy and safety of once-weekly semaglutide in uncontrolled T2DM patients who were drug-naïve or receiving anti-diabetic medications were examined in the SUSTAIN 1–5 clinical trials in 2018 [180]. All 3918 patients were randomly assigned to receive either 0.5 mg or 1 mg of semaglutide-except for SUSTAIN 3 only 1 mg was used-, comparators, or placebo [180]. It was established that once-weekly semaglutide outperformed other comparators, such as sitagliptin, exenatide, insulin glargine, and placebo, in terms of glycemic management, BMI, and weight loss [180]. For semaglutide, a weight loss of between 2.3 and 6.3 kg was reported [180]. However, the semaglutide therapy group saw more cases of nausea and vomiting. Nevertheless, the loss of weight from nausea and vomiting was quite minor, ranging from 0.07 to 0.5 kg [180].

We now go on to SUSTAIN 6, which was a multi-national, randomized, double-blind, double-placebo controlled trial with the goal of assessing the cardiovascular safety of semaglutide in T2DM patients [181]. A total of 3297 T2DM patients were randomly assigned to receive either volume-matched placebo, semaglutide 1 mg once weekly, or 0.5 mg [181]. At week 104, the average weight loss for the 0.5 mg and 1 mg semaglutide groups was 3.6 kg and 4.9 kg, respectively, as opposed to a loss of 0.7 kg and 0.5 kg in the placebo groups [181]. In SUSTAIN 7, an open label, randomized, parallel-group phase 3 trial, semaglutide and dulaglutide were compared. In total, 199 T2DM patients receiving metformin were randomly assigned to receive either 0.5 mg or 1.0 mg of semaglutide subcutaneously or 0.75 mg or 1.5 mg of dulaglutide once per week [182]. For doses of 0.5 mg semaglutide, 1 mg semaglutide, 0.75 mg dulaglutide, and 1.5 mg dulaglutide, respectively, the average weight decreases were 4.6 kg, 6.5 kg, 2.3 kg, and 3.0 kg [182]. Additionally, in SUSTAIN 8, a randomized, double-blind, parallel-group, phase 3 trial, 788 T2DM patients using metformin were randomly assigned to receive either 1.0 mg of semaglutide once a week or 300 mg of canagliflozin once a day [183]. With semaglutide and canagliflozin, the average weight loss after 52 weeks was 5.3 kg and 4.2 kg, respectively [183]. In SUSTAIN 9, a randomized, double-blinded, international trial, 302 T2DM patients on an SGLT-2 inhibitor with metformin with/without sulfonylurea were randomly assigned to receive either a placebo or 1 mg of semaglutide once per week [184]. The average weight reduction was 3.81 kg for the semaglutide group compared to 0.9 kg for the placebo group [184]. Additionally, in the SUSTAIN 10 study, semaglutide’s safety and effectiveness were contrasted with liraglutide’s when administered in T2DM patients using no more than 3 oral medications [185]. In this phase 3 randomized, open-label trial, 577 T2DM patients were randomly assigned to receive either 1.2 mg subcutaneous liraglutide once day or 1 mg subcutaneous semaglutide once weekly [185]. The average weight loss with semaglutide was larger with a value of 5.8 kg compared to only 1.9 kg with liraglutide, in addition to superior glycemic control [185]. With the exception of gastrointestinal side effects, which were more common in the semaglutide group, safety profiles were essentially equal [185]. In the SUSTAIN 11 trial, once-weekly semaglutide and insulin aspart were tested for effectiveness and safety [186]. In that multi-nation, randomized, parallel-group, phase 3 trial, 1748 patients with uncontrolled T2DM who were taking metformin and insulin glargine were randomly assigned to receive either 1 mg of subcutaneous semaglutide once a week or 100 IU of insulin aspart three times a day [186]. At 52 weeks, semaglutide induced an average weight loss of 4.1 kg, while the aspart group experienced an average weight gain of 2.8 kg [186]. The semaglutide group also saw improved glycemic results [186]. The semaglutide group experienced more adverse events, primarily gastrointestinal [186].

Moving to the second program, the Pioneer studies, which sought to determine the effectiveness and safety of oral semaglutide [187]. In PIONEER 1, a 26-week, double-blind, randomized, parallel-group, multinational, phase 3 trial, 703 patients were randomized at 1:1:1:1 to 3 mg, 7 mg, or 14 mg of once-daily oral semaglutide or placebo [187]. This study showed the effectiveness of oral semaglutide on naïve T2DM patients with lifestyle interventions [187]. When compared to placebo, patients who received 7 mg and 14 mg orally demonstrated significant body weight reductions of at least 5% [187]. It was possible to lose weight in a dose-dependent manner; the maximum oral semaglutide dosage (14 mg once daily) produced the greatest weight loss (4.1 kg) [187]. In PIONEER2, a 52-week, international, open-label, phase 3 trial, 822 individuals were randomized at 1:1 into 14 mg of oral semaglutide or 25 mg of oral empagliflozin once daily [188]. These patients had uncontrolled T2DM and were taking metformin. For oral semaglutide and empagliflozin, the weight loss at week 52 was 4.7 kg and 3.8 kg, respectively [188]. In the PIONEER 3 trial, a 78-week, randomized, double-blind, double-dummy study, 1864 individuals with TD2M who were taking metformin with or without a sulfonylurea were randomized at 1:1:1:1 to receive 3 mg, 7 mg, or 14 mg of oral semaglutide daily or 100 mg of sitagliptin daily [189]. At 26 weeks, the initial findings indicated that the daily doses of 3 mg, 7 mg, and 14 mg semaglutide and 100 mg sitagliptin, respectively, resulted in average weight reductions of 1.2 kg, 2.2 kg, 3.1 kg, and 0.6 kg [189]. In PIONEER 4, a randomized, double-blind, phase 3 trial, 711 patients were randomly assigned at a ratio of 2:2:1 to receive 14 mg of oral semaglutide once daily, 3 mg of subcutaneous liraglutide once daily, or placebo [190]. In this research, oral semaglutide was compared to another GLP-1 receptor agonist, liraglutide, in T2DM patients taking metformin with/without SGLT-2 inhibitor [190]. For semaglutide, liraglutide, and placebo, the average weight loss at week 26 was 4.4 kg, 3.1 kg, and 0.5 kg, respectively [190]. In PIONEER 5, a randomized, double-blind, phase 3 trial where 324 patients were randomized into 1:1 of 14 mg oral semaglutide or placebo, oral semaglutide was evaluated in T2DM patients with renal impairment on sulfonylurea, metformin, or both, or basal insulin with/without metformin [191]. In comparison to placebo, semaglutide caused average weight decreases of 3.4 kg as opposed to only 0.9 kg in the placebo group [191]. Semaglutide was discovered to have a similar renal safety profile to other GLP-1 receptor agonist drugs [191]. In PIONEER 6, a randomized, double-blind, phase 3 trial, 3183 individuals with T2DM with cardiovascular risks were randomly assigned to receive 14 mg of oral semaglutide or a placebo [192]. This study examined the cardiovascular safety of oral semaglutide. Findings have demonstrated that 14 mg of oral semaglutide did not raise cardiovascular risks and caused an average weight loss of 4.2 kg as opposed to a placebo’s 0.8 kg [192]. The clinical trial PIONEER 7, a 52-week, randomized, open-label, phase 3 trial in which 504 T2DM patients on metformin with/without sulfonylurea were randomized into 1:1 flexible dose adjustment of 3 mg, 7 mg, or 14 mg of oral semaglutide once daily or sitagliptin 100 mg once daily, examined the effectiveness and safety of oral semaglutide dosing flexibility [193,194]. In comparison to the sitagliptin group, the semaglutide group experienced an average weight loss from baseline of 2.4 kg as opposed to 0.9 kg [193,194]. In PIONEER 8, a 52-week, randomized, double-blind, phase 3 trial, 731 T2DM patients were randomly assigned to receive placebo, 3 mg, 7 mg, or 14 mg of oral semaglutide daily, or to receive either placebo or insulin with or without metformin [195]. With oral semaglutide doses of 3 mg, 7 mg, and 14 mg, respectively, the average weight loss in the semaglutide group was 1.4 kg, 2.4 kg, and 3.7 kg, as opposed to 0.4 kg in the placebo group [195]. Additionally, uncontrolled T2DM patients who are taking one oral antidiabetic medication or on lifestyle treatments were included in PIONEER 9, a 52-week, randomized, phase 2/3 trial conducted in Japan [196]. A total of 248 individuals were randomized at 1:1:1:1:1 of 3 mg, 7 mg, or 14 mg oral semaglutide or placebo, or to an open-label 0.9 mg subcutaneous liraglutide once-daily [196]. At week 52, average weight losses for semaglutide 14 mg oral were larger than those for placebo (*p* = 0.0019) and liraglutide (*p* < 0.0001), which were both statistically significant [196]. Finally, PIONEER 10—a randomized, open-label, active-controlled, phase 3 trial in which 458 T2DM patients were randomly assigned to receive either 3 mg, 7 mg, or 14 mg of oral semaglutide once daily or 0.75 mg of subcutaneous dulaglutide once a week—was used to compare oral semaglutide to dulaglutide [197]. In comparison to the dulaglutide group, average weight loss after 52 weeks was 0.0 kg, 0.9 kg, and 1.6 kg for 3 mg, 7 mg, and 14 mg oral semaglutide, respectively, compared to 1.0 kg in the dulaglutide group (*p* < 0.0001 for 14 mg oral semaglutide vs. dulaglutide) [197].

Next, we will look at the STEP program, which examined semaglutide’s safety and effectiveness as an anti-obesity drug [198,199,200]. In STEP 1, a double-blind, randomized phase 3 trial in which 1961 adults with BMI 30 or 27 and weight-related co-morbidities excluding diabetes were randomized into a 2:1 ratio of 2.4 mg subcutaneous semaglutide once weekly or placebo for 68 weeks, the effectiveness of semaglutide in reducing weight in addition to lifestyle modifications was examined [198]. At 68 weeks, the average weight loss for semaglutide and placebo was 15.3 kg and 2.6 kg, respectively [198]. In STEP 2, a double-blind, double-dummy, phase 3 trial, 1210 T2DM patients with BMI 27 were randomized at 1:1:1 of 2.4 mg, 1.0 mg subcutaneous semaglutide once weekly, or placebo [199]. This study examined the weight-reduction effects of semaglutide in these individuals. Body weight decreases from baseline at 68 weeks were 9.64%, 6.99%, and 3.42% for semaglutide dosages of 2.4 mg, 1.0 mg, and placebo, respectively [199].

In STEP 3, a 68-week, double-blind, randomized, parallel-group phase 3 trial in which 611 patients were randomized into 2:1 of 2.4 mg subcutaneous semaglutide once-weekly or placebo, weight loss by semaglutide was also evaluated in overweight or obese adults [200]. Body weight decreased by 16.0% when semaglutide was used versus 5.7% when a placebo was used [200]. In STEP 4, a 68-week, double-blind, randomized, withdrawal study, 902 participants were randomly assigned to continue taking 2.4 mg of semaglutide or to receive a placebo at a 2:1 ratio for the following 48 weeks [201]. This treatment period involved 902 overweight or obese participants with a BMI of 30 or higher or participants with weight-related co-morbidities other than diabetes [201]. Semaglutide caused a 7.9 kg weight loss from weeks 20 to 68 after 68 weeks, whereas those who switched to placebo experienced a 6.9 kg weight gain [201]. Weight loss in the semaglutide group peaked between weeks 60 and 68 and has since decreased by a total of 17.4%, compared to a total weight loss of 5% in the placebo group [201]. In STEP 5, a 104-week randomized, parallel-arm phase 3 trial in which 304 overweight/obese patients with weight-related co-morbidities other than diabetes were randomly assigned to receive either 2.4 mg of semaglutide once weekly or a placebo, the safety and efficacy of long-term semaglutide use were investigated [202]. In the semaglutide group, body weight decreased by 15.2%, compared to 2.6% in the placebo group [202]. It is important to note that weight loss peaks at week 60, although the reduction was maintained for the whole 104 weeks [202].

In addition, STEP 6—a randomized, double-dummy, double-blind, placebo-controlled, phase 3 trial in which 401 participants were randomly assigned to receive either once-weekly 2.4 mg semaglutide (*n* = 199), once-weekly 1.7 mg semaglutide (*n* = 101), or placebo (*n* = 101)—studied the weight loss effects of semaglutide in overweight/obese participants of Asian ethnicity [203]. For 2.4 mg, 1.7 mg semaglutide, and placebo, the average weight loss from baseline was 13.2%, 9.6%, and 2.1%, respectively [203]. The STEP 7 trial, which randomized overweight/obese participants with or without T2DM to receive 2.4 mg subcutaneous semaglutide once weekly or a placebo, is finished, although its results have not yet been published [204]. Similarly, in STEP 8, a 68-week, open-label, randomized, phase 3 trial [205], the effects of 2.4 mg semaglutide and 3.0 mg liraglutide on weight loss were contrasted. In total, 338 patients with a BMI of 30 or lower with weight-related co-morbidities other than diabetes were randomly assigned to receive either placebo (*n* = 85), once-weekly doses of 2.4 mg semaglutide, once-daily doses of 3 mg liraglutide, or both [205]. With semaglutide, liraglutide, and the combined placebo, the average weight loss from baseline was 15.8%, 6.4%, and 1.9%, respectively [205]. The last study is STEP-TEENS, a phase 3 trial that randomly assigned 201 obese/overweight adolescents between the ages of 12 and 17 to receive either 2.4 mg of semaglutide once a week or a placebo, along with lifestyle changes [206]. In the semaglutide group, the mean change in BMI from baseline to week 68 was a decrease of 16.1% compared to a rise of 0.6% in the placebo group [206]. In clinical trials for T2DM, semaglutide had shown greater effects on weight loss compared to all comparators (SUSTAIN & PIONEER). In addition, a new dosage of 2.4 mg is now licensed as an anti-obesity medicine as a result of the STEP program’s exceptional weight loss results in non-diabetic patients. The most frequent reason for cessation is gastrointestinal adverse effects. However, compared to tirzepatide, it has less of an impact on weight loss as it was discussed below.

##### Tirzepatide

The most recent T2DM medication, tirzepatide, is twincretin with 39 amino acid synthetic peptide, a dual agonist of the GIP and GLP-1 receptors [207]. On 13 May 2022, it got FDA approval under the brand name Mounjaro^®^ [207,208]. As a subcutaneous injection, it is given once per week [207]. Tirzepatide has been shown to be effective in the treatment of non-alcoholic steatohepatitis, T2DM, and overweight and obese patients independent of their diabetes status [207,209]. Clinical trials known as the SURPASS clinical trials were used to examine the efficacy and safety of tirzepatide [209,210,211,212,213]. The tirzepatide weight loss effects from the SURPASS clinical trials are summarized in this study [210,211,212,213]. A total of 478 T2DM patients who had never taken medication were randomly assigned to receive placebo or one of three doses of tirzepatide, either 5, 10, or 15 mg, administered subcutaneously once a week in a ratio of 1:1:1:1 during the 40-week SURPASS-1 clinical trial [210]. According to the dose, tirzepatide caused an average weight reduction of between 7.0 and 9.5 kg [210]. In SURPASS-2, an open-labeled phase 3 trial, tirzepatide and semaglutide were compared [211]. In total, 1879 T2DM patients on only metformin were enrolled in the research [211]. Jn total, five, 10, and 15 milligrams of tirzepatide and a weekly dose of semaglutide of 1 milligram each were randomly assigned to patients [211]. Tirzepatide outperformed semaglutide at week 40, causing average weight losses of 7.6 kg, 9.3 kg, and 11.2 kg at doses of 5 mg, 10 mg, and 15 mg, respectively [211]. However semaglutide caused a 5.7 kg weight decrease on average [211]. In SURPASS-3, an open-label, phase 3 trial, tirzepatide and insulin degludec were evaluated in T2DM patients using metformin with or without an SGLT-2 inhibitor [212]. A total of 1444 T2DM patients were randomly assigned (1:1:1:1) to receive either insulin degludec once daily or tirzepatide 5 mg, 10 mg, or 15 mg once weekly [212]. Weight loss after 52 weeks of treatment in the tirzepatide group was directly correlated with dose, with average losses of 7.5 kg, 10.7 kg, and 12.9 kg, respectively, as opposed to gains of 2.3 kg in the degludec group [212]. Tirzepatide and insulin glargine were compared in the 2002 T2DM patients with cardiovascular risk, taking metformin with/without a sulfonylurea or SGLT-2 inhibitor in the open-labeled, phase 3 parallel-group research SURPASS-4 [213]. After 104 weeks, the average weight reduction for 5 mg (*n* = 329), 10 mg (*n* = 328), and 15 mg (*n* = 338) tirzepatide, respectively, was 7.1 kg, 9.5 kg, and 11.7 kg vs. a gain of 1.9 kg on average in the glargine group (*n* = 1000 patients) [213]. When compared to insulin glargine, tirzepatide had a similar rate of cardiovascular events [213]. Tirzepatide was added to insulin glargine patients with or without metformin in SURPASS-5, a double-blinded, randomized, phase 3 experiment [214]. In total, 475 patients in all were randomly assigned to receive placebo or 5 mg, 10 mg, or 15 mg of tirzepatide once weekly [214]. In comparison to the placebo group, which gained 1.7 kg on average, the tirzepatide group had lost an average of 10.9 kg [214]. Two of the exclusive tirzepatide clinical studies that were carried out specifically for Asian markets revealed their findings [215,216]. The first was SURPASS J-mono, a double-blinded study in which patients with type 2 diabetes were either drug-naïve or had stopped taking monotherapy before the trial started [215]. In total, 636 patients in total were randomly assigned to receive either 0.75 mg of dulaglutide or 5 mg, 10 mg, or 15 mg of tirzepatide once weekly [215]. The average weight loss in the tirzepatide groups was 5.8 kg, 8.5 kg, and 10.7 kg for doses of 5 mg, 10 mg, and 15 mg of tirzepatide, respectively [215]. Dulaglutide, however, only caused an average weight loss of 0.5 kg [215]. The second published study SURPASS J-combo; an open-labeled, randomized trial was created to monitor long-term adverse events [216]. Throughout the course of 52 weeks, 5 mg, 10 mg, or 15 mg of tirzepatide were administered to 443 individuals in addition to a non-incretin anti-diabetic medication [216]. A weight loss was seen, with mean average reductions from baseline of 3.8 kg, 7.5 kg, and 10.2 kg in the 5 mg, 10 mg, and 15 mg tirzepatide groups, respectively [216]. The most common reported side effects were nasopharyngitis, nausea, constipation, diarrhea, and decreased appetite [216]. Tirizepatide’s anti-obesity properties were highlighted by the SURMOUNT clinical study series [217]. The SURMOUNT-1 study is the only one whose results have been published; the others are either ongoing or have not done so [217]. A total of 2539 patients with a BMI of 30 kg/m^2^ or higher, or a BMI of 27 kg/m^2^ or higher with at least one obesity-related health issue other than diabetes, were randomly assigned to receive tirzepatide in doses of 5 mg, 10 mg, or 15 mg daily for 72 weeks, or a placebo, in the SURMOUNT-1 double-blinded, randomized, phase 3 trial [217]. For the dosages of 5 mg, 10 mg, 15 mg tirzepatide, and placebo, the percentage of participants who lost 5% or more of their body weight was 85%, 89%, 91%, and 35%, respectively [217]. The astounding results had showed that at the 10 mg and 15 mg tirzepatide doses, respectively, 50% and 57% of subjects had achieved body weight reductions of 20% or more [217]. However, the placebo group only experienced a 3% decrease in body weight [217]. The most frequent side effects, particularly during dose escalation, were mild to moderate gastrointestinal problems [217]. The SURMOUNT-1 study’s encouraging outcomes signal a paradigm shift and provide fresh insights into the management of obesity [217]. Several research on tirzepatide are currently continuing, and some, like SURPASS-6, SURPASS-CVOT, SURPASS-AP-Combo, SURPASS-PEDS, SUMMIT, SURMOUNT-CN, SURMOUNT-2, SURMOUNT-3, and SURMOUNT-4, have not yet published their findings [218,219,220]. Only the SURMOUNT clinical trials will shed light on tirzepatide’s anti-obesity properties [218]. Overall, the newest T2DM medication, tirzepatide, is the twincretin molecule that has demonstrated astounding weight loss results and outperformed all other comparable drugs, including semaglutide. Despite this, more research is needed to determine its long-term efficacy and safety.

The potential of biguanides, alpha-glucosidase inhibitors, DPP-4inhibitors, SGLT-2 inhibitors, and GLP-1 receptor agonists as anti-obesity medications was highlighted in this study’s evaluation of many clinical trials that looked at antihyperglycemic treatments that have demonstrated weight loss effects. Look at Table 1.

## 4. Summary, Conclusions, and Future Directions

Because it increases the risk of several diseases, obesity is a multifactorial metabolic disorder that can lower quality of life and even result in mortality. Over the past 50 years, the prevalence of obesity has increased to epidemic levels worldwide. Clinical research has demonstrated that, in addition to a number of obesity management strategies, some antihyperglycemic medications have a weight loss impact, while others have a gain or neutral effect. The potential of biguanides, alpha-glucosidase inhibitors, DPP-4inhibitors, SGLT-2 inhibitors, and GLP-1 receptor agonists as anti-obesity medications were highlighted in this study’s evaluation of many clinical trials that looked at antihyperglycemic treatments that have demonstrated weight loss effects. See Table 1.

In conclusion, DPP-4 has a neutral effect on weight or a minor reduction in weight, acarbose has mild effects, metformin and SGLT-2 inhibitors have moderate effects on weight loss, and some of the GLP-1 agonist drugs have the strongest and most promising results as anti-obesity drugs. The newest T2DM medicine, tirzepatide, outperformed all other comparators, including semaglutide, in terms of weight loss effects. Despite this, more research is needed to determine its long-term efficacy and safety.

Further studies are required to describe the structural activity relationship and molecular mechanism behind the anti-obesity property of T2DM medications and the molecular modeling aspect correlating the weight loss effects of T2DM agents at the molecular level. For example, numerous studies suggested that protein tyrosine phosphatase 1B (PTP1B) inhibitors could be a promising treatment for T2DM and obesity [221,222]. Moreover, it has been demonstrated that liraglutide downregulated PTP1B which could explain its anti-diabetics and anti-obesity effects [222]. Therefore, studying if there are any inhibitor effects for GLP-1 receptor agonists on PTP1B will be worthwhile. In addition, it has been suggested that AMP-activated protein kinase (AMPK) may be a target for the treatment of obesity and diabetes [223]. Metformin effects on AMPK has a crucial role in determining its anti-diabetic properties and may also play a role in its anti-obesity effects [223,224]. However, some studies have demonstrated that these effects are also mediated by AMPK-independent pathways such as increasing fibroblast growth factor 21 (FGF21) expression [225]. Future research should focus on examining all possible molecular mechanisms and structural activity relationships that identify the weight reduction property of T2DM agents.

## Figures and Tables

**Figure 1 life-13-01012-f001:**
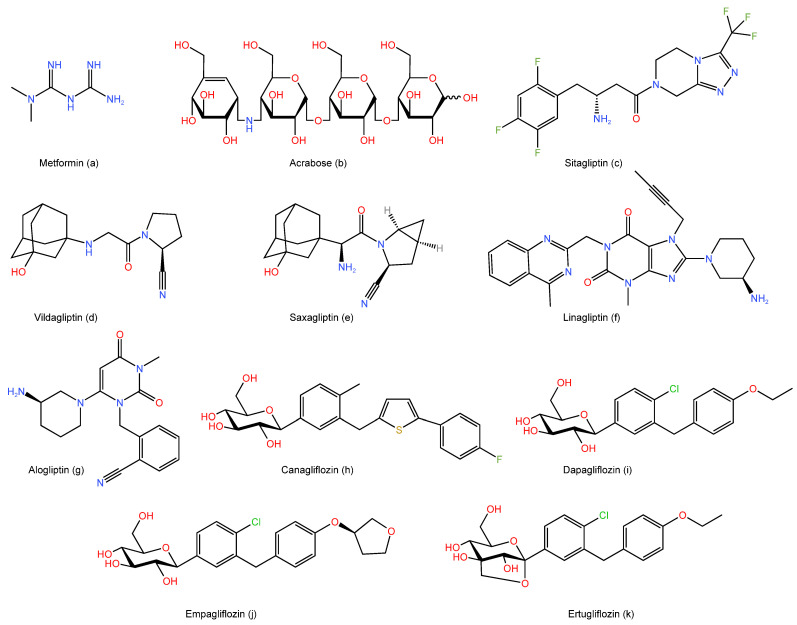
The chemical structures of: (**a**) Metformin, (**b**) Acarbose, (**c**) Sitagliptin, (**d**) Vildagliptin, (**e**) Saxagliptin, (**f**) Linagliptin, (**g**) Alogliptin, (**h**) Canagliflozin, (**i**) Dapagliflozin, (**j**) Empagliflozin, and (**k**) Ertugliflozin.

**Figure 2 life-13-01012-f002:**
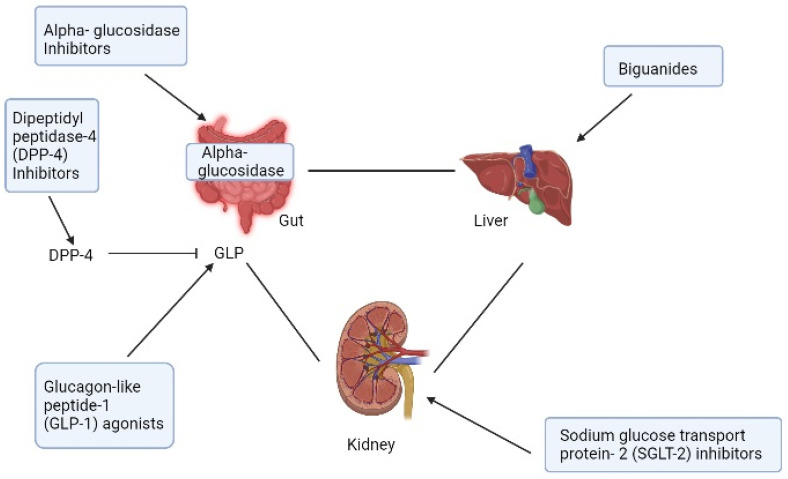
The primary mechanism of action of the biguanide, alpha-glucosidase inhibitors, dipeptidyl peptidase-4 (DPP-4) inhibitors, sodium-dependent glucose cotransporter proteins-2 (SGLT-2) inhibitors, and glucagon-like peptide-1 (GLP-1) agonists.

**Table 1 life-13-01012-t001:** Summary of the effects of the selected anti-diabetic medications on a weight loss with number of the studies and subjects reviewed in this research for each drug.

Agent	Class	Total Number of Studies Reviewed	Total Number of Subjects (Characteristic of Subjects)	Weight Loss Effect
Metformin	Biguanide	Four clinical studies	*n* = 3627 (T2DM patients)	Moderate
		One systematic review and meta-analysis	*n* = 743 (individuals with schizophrenia or schizoaffective disorder)	
		One systemic review	*n* = 1004 (T2DM patients)	
Acarbose	Alpha-Glucosidase Inhibitors	Three clinical studies	*n* = 68,809 (T2DM subjects)	Mild
		One systematic review and meta-analysis	*n* = 11,877 (T2DM subjects)	
		One systematic review and meta-analysis	*n* = 164 (non-diabetic overweight and obese subjects)	
Sitagliptin	Dipeptidyl peptidase 4 (DPP-4) inhibitor	Three clinical studies	*n* = 2012 (T2DM subjects)	Mild
		One clinical study	*n* = 24 (Prediabetics subjects)	
Vildagliptin	DPP-4 inhibitor	One clinical study	*n* = 2340 (T2DM patients)	Neutral
		One systematic review	*n* = 741 (T2DM patients)	
		One meta-analysis	*n* = 9274 (T2DM patients)	
Saxagliptin	DPP-4 inhibitor	Three clinical studies	*n* = 2012 (T2DM subjects)	Neutral
		One clinical study	*n* = 24 (Prediabetics subjects)	
Linagliptin	DPP-4 inhibitor	Four clinical studies	*n* = 487 (T2DM patients)	Neutral
Alogliptin	DPP-4 inhibitor	Three clinical studies	*n* = 2822 (T2DM patients)	There was a discrepancy in alogliptin effects on body weight so further studies are needed to investigate its effect on weight.
		One comprehensive review and meta-analysis	*n* = 1887 (T2DM patients)	
Canagliflozin	Sodium-dependent glucose cotransporter proteins-2 (SGLT-2) inhibitors	Three clinical studies	*n* = 2079 (T2DM subjects)	Moderate
		one systemic review	*n* = 13,158 (T2DM subjects)	
Dapagliflozin	SGLT-2 inhibitors	Three clinical studies	*n* = 17,442 (T2DM subjects)	Moderate
		one clinical study	*n* = 50 (obese adults with prediabetes)	
Empagliflozin	SGLT-2 inhibitors	Three clinical studies	*n* = 4482 (T2DM subjects)	Moderate
Ertugliflozin	SGLT-2 inhibitors	Four clinical studies	*n* = 2779 (T2DM subjects)	Moderate
Exenatide	Glucagon-like peptide-1 (GLP-1) receptor agonist	Nine clinical studies	*n* = 4430 (T2DM patients)	Moderate
		One clinical trial	*n* = 182 (non-diabetic overweight/obese women)	
		One meta-analysis	*n* = 362 (obese non-diabetic)	
Liraglutide	GLP-1 receptor agonist	Seven clinical studies	*n* = 3810 (T2DM patients)	Strong
		Two clinical studies	*n* = 1242 (obeseT2DM patients)	
		One systemic review and meta-analysis	*n* = 4754 (obese and/T2DM patients)	It has been granted FDA approval for the management of obesity.
		One meta-analysis	*n* = 8060 (overweight adults)	
Dulaglutide	GLP-1 receptor agonist	Twelve clinical studies	*n* = 8151 (T2DM patients)	Mild to moderate effect that were proportional to the dose used.
Lixisenatide	GLP-1 receptor agonist	Thirteen clinical studies	*n* = 5495 (T2DM patients)	Mild
Semaglutide (subcutaneous)	GLP-1 receptor agonist	Six clinical studies	*n* = 6911 (T2DM patients)	Strong
		Seven clinical studies	*n* = 4718 (obese/overweight patients)	
		Systemic review	*n* = 3918 (T2DM patients)	It has been granted FDA approval for the management of obesity.
		One clinical study	*n* = 1210 (obese and/T2DM patients)	Moderate to strong weight loss effects.
Oral semaglutide	GLP-1 receptor agonist	Ten clinical studies	*n* = 9548 (T2DM patients)	It has been granted FDA approval for the management of obesity.
Tirzepatide	Dual gastric inhibitory peptide (GIP) and GLP-1 receptor agonist (twincretin).	Seven clinical studies	*n* = 7357 (T2DM patients)	Tirzepatide, the newest T2DM medication, has shown the strongest weight loss effects and was superior to all comparators including semaglutide. Despite this, further studies are warranted to examine its long-term safety and efficacy.
		One clinical study	*n* = 2539 (obese/overweight patients)	

## Data Availability

Not applicable.

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
