# Peer review of "A Comprehensive Review on Weight Loss Associated with Anti-Diabetic Medications"

_life, 2023, doi:10.3390/life13041012_

Round 1

Reviewer 1 Report

A review manuscript of Haddad et al focussed towards correletive study of the effects of Anti-diabetic medications on weight loss. This was supported by sufficient literature. Followingcomments are to be addressed in the revision.

1. In the introduction, the recent WHO report to be cited for reference 1 and the text is to be updated, accordingly.

2. In Page-10; line-367, under 2.3.3. Saxagliptin: pls clarify-In 1282.

3. In Page-17; line-632, under 2.5.1. Exenatide: pls clarify-DURATION. What it refers to? Similarly, DURATION-NEO-1,2...

4. The text 'The potential of biguanides, alpha-glucosidase inhibitors, -------------------. Look at Table 1.' This text and Table 1 to be moved before the conclusion section. 

5. Conclusion should be based on your own conclusions from the literature review, and how do you think this direction would go further. 

6. All the agents mentioned in table 1 must be represented by their chemical structures in Figure 1. 

7. One paragraph describing the logical reasons including structural requirement, SAR studies, etc.. behind the anti-obesity property of antidiabetic agents.

8. One section to be included as "computational aspects of antidiabetic drug design in correlation to anti-obesity property; This one must describe the molecular modeling aspect correlating the weight reduction property of antidiabetic drugs at the molecular level through in silico computational research papers.

Author Response

Reviewer 1

We thank the reviewer for his fruitful comments which have been taken into consideration and addressed accordingly:

  • Comment:  In the introduction, the recent WHO report to be cited for reference 1 and the text is to be updated, accordingly.
  • Response: A reference has been added and the text has been updated, accordingly
  • Comment: In Page-10; line-367, under 2.3.3. Saxagliptin: pls clarify-In 1282.
  • Response: This sentence has been rephrased.

  • Comment:  In Page-17; line-632, under 2.5.1. Exenatide: pls clarify-DURATION. What it refers to? Similarly, DURATION-NEO-1,2...
  • Response: The meaning of these abbreviations has been added and modified in the text.

DURATION: Diabetes therapy Utilization: Researching changes in A1C, weight and other factors

DURATION-Neo-1: DURATION-Neoadjuvant-1

  • Comment: The text 'The potential of biguanides, alpha-glucosidase inhibitors, -------------------. Look at Table 1.' This text and Table 1 to be moved before the conclusion section.
  • Response: This has been modified and moved before the conclusion section.

  • Comment: Conclusion should be based on your own conclusions from the literature review, and how do you think this direction would go further. 
  • Response: The conclusion has been modified accordingly.
  • Comment: All the agents mentioned in table 1 must be represented by their chemical structures in Figure 1. 

  • Response: The chemical structures of all small molecule drugs mentioned in this review have been added; however, since GLP-1 receptor agonists are peptides, we have included their textual descriptions as well. For instance, Exendin-4, a 39 amino acid compound with 53% similarity to human GLP-1, and Dulaglutide, a DPP-IV-protected GLP-1 analogue covalently linked to a human IgG4-Fc heavy chain.

  • Comment: One paragraph describing the logical reasons including structural requirement, SAR studies, etc.. behind the anti-obesity property of antidiabetic agents.

Response: A paragraph mentioning this topic has been included before the conclusion because it is one that is very intriguing to talk about. (pages 37-38, lines 1321-1336, before accepting the track changes)

  • Comment: One section to be included as "computational aspects of antidiabetic drug design in correlation to anti-obesity property; This one must describe the molecular modeling aspect correlating the weight reduction property of antidiabetic drugs at the molecular level through in silico computational research papers.
  • Response: A paragraph has been added before the conclusion, and we provided examples of potential molecular mechanisms for these effects. However, since there aren't many studies or reviews that discuss these to date, future research should concentrate on examining all potential molecular mechanisms and structural activity relationships that identify the weight reduction property of T2DM agents. (pages 37-38, lines 1321-1336, before accepting the track changes)

Reviewer 2 Report

Thank you for inviting me to read this interesting and very relevant paper. The purpose of this paper is to present a narrative summary of antihyperglycemic medication effect on weight loss.

Overall I think this is an interesting paper and hope one day can become a systematic review with meta-analysis. For now, agree that is important to establish the effects of these medications on weight reduction in populations with and without type 2 diabetes, with obesity (BMI >30kg/m2)

Major Comments

1) Ideally a methods section detailing paper selection criteria, definitions for "mild weight loss" and "Moderate weight loss", population characterisations (BMI and Health status). 

2) Describing the paper in the title as "efficacy" of anti-diabetic medication in weight loss, while it seems appropriate, I question how this narrative synthesis of the literature can meet the accepted scientific definition of efficacy? maybe it's more of a change in weight associated with use of anti-diabetes medication in those with or without diabetes or with or without obesity.? something to consider thank you.

3) it would be more systematic to provide, for each drug a) number or papers reviewed using this drug, the description of the population, intervention and outcomes (i.e. change in weight over time). 

Minor Comments: 

1)  Language and reference to patients as "Sufferers" needs review. This is currently a move to avoid terms that imply blame or burden of disease such as in the term "sufferers"  Pg 2 line 49. More appropriate line here would be "... affects more than 9% of individuals living with diabetes mellitus".

2) Pg 2 line 51 "high calorie diets" suggest changing to "poor dietary choices" as it covers of on energy density and low nutrient quality foods.

3) Pg 2 Line 67. "Diet plans and increased physical activity are not improvements in lifestyle" as referenced in this text.  Noting, these are tools to help improve quality of lifestyle. Suggest rephrasing to "improvements in lifestyle habits, including strategies to improve diet and exercise quality".

Author Response

Reviewer 2

We thank the reviewer for his valuable comments which have been taken into consideration and addressed accordingly:

Major comments:

  • Comment: 1) Ideally a methods section detailing paper selection criteria, definitions for "mild weight loss" and "Moderate weight loss", population characterisations (BMI and Health status). 
  • Response: A method section detailing all the required information has been added.
  • Comment: 2) Describing the paper in the title as "efficacy" of anti-diabetic medication in weight loss, while it seems appropriate, I question how this narrative synthesis of the literature can meet the accepted scientific definition of efficacy? maybe it's more of a change in weight associated with use of anti-diabetes medication in those with or without diabetes or with or without obesity.? something to consider thank you.
  • Response: The title has been modified accordingly. Thank you for your suggestion.
  • Comment: 3) it would be more systematic to provide, for each drug a) number or papers reviewed using this drug, the description of the population, intervention and outcomes (i.e. change in weight over time). 
  •  
  • Response: the number of trials analyzed for each medicine has been added to Table 1, and the review's whole text has been examined for any omissions of information about the population, the intervention, and the results.

Minor Comments: 

  • Comment: Language and reference to patients as "Sufferers" needs review. This is currently a move to avoid terms that imply blame or burden of disease such as in the term "sufferers"  Pg 2 line 49. More appropriate line here would be "... affects more than 9% ofindividuals living with diabetes mellitus".
  • Response: This has been modified.
  • Comment 2) Pg 2 line 51 "high calorie diets" suggest changing to "poor dietary choices" as it covers of on energy density and low nutrient quality foods.
  • Response: This has been modified.
  • Comment 3) Pg 2 Line 67. "Diet plans and increased physical activity are not improvements in lifestyle" as referenced in this text.  Noting, these are tools to help improve quality of lifestyle. Suggest rephrasing to "improvements in lifestyle habits, including strategies to improve diet and exercise quality".
  • Response: This has been modified.

Reviewer 3 Report

The review by Haddad et al talks about the well known medications that have an accepted place and well established role as anti obesity medications and are used in the treatment and management of Obesity since the last decade and more. The review does not add anything new to the already existing knowledge.  I am not sure why the authors consider them at the present time as anti diabetics. The literature is abound with reviews of the same. Can the authors mention what is new about their review.

A major issue with the paper is that there is no methodology of how the review was carried out.

For the purpose of a review, the Table 1 provided by the authors is very minimal and needs to be more extensive, mentioning the different experimental details, sample size and the specific results rather than saying , minimal weight loss.

Author Response

Reviewer 3

We thank the reviewer for his fruitful comments which have been taken into consideration and addressed accordingly:

  • Comment: Can the authors mention what is new about their review:
  • Response: In this study, we addressed all currently available molecules and included the most published studies that reported weight data for T2DM medications. For instance, although not yet having FDA clearance for the treatment of obesity, the newest T2DM, tirzepatide, which has received FDA approval on May 13, 2022, has demonstrated the strongest effects on weight loss and outperformed all comparators, including semaglutide. The majority of the research we looked at for this agent's effects on weight were completed between 2021 and 2023. Additionally, despite semaglutide being approved in 2017 for the treatment of T2DM, it wasn't until 2021 that it received permission for use as an anti-obesity medication. As a result, the majority of the most widely disseminated research on the impact of T2DM medications on weight is included in this review.
  • A major issue with the paper is that there is no methodology of how the review was carried out.
  • Response: There is now a method section with all the necessary information.
  • For the purpose of a review, the Table 1 provided by the authors is very minimal and needs to be more extensive, mentioning the different experimental details, sample size and the specific results rather than saying , minimal weight loss.
  • Response: The necessary information has been added to Table 1 in the following ways: • Regarding the mild and moderate impacts that we described in the table, we added the definition of mild, moderate, and strong effects in the method section.

Reviewer 4 Report

Karaman et al. reviewed the literature broadly to clarify the role of anti-diabetic medications on weight loss.

The authors reviewed most published studies reporting weight data and discussed all currently available molecules. The introduction set the stage adequately.

The text does not mention how the authors conducted the literature search. On the other hand, this is tolerable since it is not a systematic review.

The text has some typos. In addition, I would suggest slimming down the table in the weight effect column. For example, I would replace "It has a mild weight loss effect " with "mild."

I recommend accepting the paper after all.

Author Response

Reviewer 4

We thank the reviewer for his kind comments which have been taken into consideration and addressed accordingly:

  • Comment: The authors reviewed most published studies reporting weight data and discussed all currently available molecules. The introduction set the stage adequately.
  • Response: Thank you for your kind comments.
  • The text does not mention how the authors conducted the literature search. On the other hand, this is tolerable since it is not a systematic review.
  • Response: A method section has been added detailing with all the required information.
  • Comment: The text has some typos.
  • Response: This has been modified
  • Comment: In addition, I would suggest slimming down the table in the weight effect column. For example, I would replace "It has a mild weight loss effect " with "mild."
  • Response: This has been modified

Round 2

Reviewer 1 Report

Necessary corrections were done by authors.

Reviewer 2 Report

Thank you for making the suggested changes and considering comments made. The revised manuscript is an improvement, thank you for your time.

Reviewer 3 Report

The authors have taken an effort to improve their manuscript. The suggested changes considering the comments have been made. The revised manuscript is an improvement and the modified title is more apt than the earlier version. All the best.